# Asymmetric cognitive learning mechanisms underlying the persistence of intergroup bias
Orit Nafcha [1,2,3] ✉ & Uri Hertz [3,4] ✉

Intergroup bias, the tendency to favor ingroups and be hostile towards outgroups, underlies many societal problems and persists even when intergroup members interact and share experiences. Here we study the way cognitive learning processes contribute to the persistence of intergroup bias. Participants played a game with ingroup and outgroup bot-players that entailed collecting stars and could sacrifice a move to zap another player. We found that intergroup bias persisted as participants were more likely to zap outgroup players, regardless of their zapping behavior. Using a computational model, we found that this bias was caused by asymmetries in three learning mechanisms. Participants had a greater prior bias to zap out-group players, they learned more readily about the negative behavior of out-groups and were less likely to attribute the positive behavior of one out-group player to other out-group players. Our results uncover the way cognitive social learning mechanisms shape and confound intergroup dynamics.

Humans are social creatures who have evolved to live in groups[1–3]. From an early age, we are wired to seek out social connections and interactions with others[4,5]. Belonging to a group confers many advantages, from social learning to emotional support and sharing in efforts to amass resources and raise one's offspring[6]. Ingroup affiliation is also defined in relation to outgroup members in that a sense of 'us' is usually accompanied by a sense of 'them', which may result in ingroup favoritism and intergroup conflict[7,8]. Recent research on resolution of intergroup conflict highlights the importance of shared reality and interaction between members of different groups in diffusing intergroup tension and overcome stereotypes[9]. Yet although many societies are made up of people from diverse cultural backgrounds who work together and interact with one another, intergroup tensions and stereotypes persist. Here we empirically examine how such persistence can be explained from a cognitive learning perspective.

Group identity is created through a process of social categorization in which people form their sense of self-concept based on their membership and classification into a social group[10–12]. This categorization into groups may be based on shared characteristics such as race, ethnicity, gender, religion and nationality[11,13,14]. Yet research shows that even assignment to random groups based on an arbitrary criterion such as color or task performance (i.e., minimal group procedure[15,16]) is sufficient to evoke intragroup identity and intergroup discrimination[16–18]. Social categorization facilitates simplification of the social world and generalization of existing knowledge about certain groups to new group members[10]. Once individuals classify themselves into a group, they begin to develop a sense of group belonging, loyalty, and similarity[19–21]. This identity is characterized by a distinct set of beliefs, values, and norms that differentiate the group from others[11]. Group identity influences one's thoughts, beliefs, feelings, and social behavior toward other ingroup partners and toward outgroup members and can also lead to the activation and application of inaccurate stereotypes and prejudices[11,22–24].

Intergroup bias, also known as ingroup favoritism or ingroup love[13,25], is defined as the tendency for individuals to favor members of their own group and show more positive attitudes and behaviors toward them[8,26]. Ingroup favoritism can have both positive and negative consequences. On the one hand, it can lead to increased positive attributions and attraction[27,28], cooperation[8,29], trust[30,31], and social cohesion within the group[32]. On the other hand, ingroup favoritism can find expression in negative evaluations and discrimination toward outgroup members[13]. Outgroup bias, also known as outgroup hate, refers to the tendency for individuals to hold negative attitudes and beliefs toward individuals belonging to a distinctly different group. Expressions of this type of hostility include prejudice, discrimination, dehumanization, and even violence[25,33–36]. Among the

[1]School of Psychological Sciences, University of Haifa, Haifa 3498838, Israel. [2]Translational Neuromodeling Unit (TNU), Institute for Biomedical Engineering, University of Zurich & ETH Zurich, Zürich 8032, Switzerland. [3]The Institute of Information Processing and Decision Making (IIPDM), University of Haifa, Haifa 3498838, Israel. [4]Department of Cognitive Sciences, University of Haifa, Haifa 3498838, Israel. ✉e-mail: ornafcha@gmail.com; uhertz@cog.haifa.ac.il

motivating factors for this bias are perceived competition for resources, differences in beliefs or values, or perceived threat to one's social identity.

The persistence of these intragroup and intergroup biases in globalized, multinational, and multicultural modern societies in which people have opportunities to experience and interact with outgroup individuals is the source of many conflicts and tensions. For example, people may tend to favor ingroups as neighbors, employees and service providers[37,38], leading to exclusion and discrimination of outgroup members, which can have detrimental effects on society and individuals. Reducing intergroup bias has been the topic of many scientific investigations. One important direction suggested by intergroup contact theory[39,40] is that increasing the contact and shared experiences between members of different groups can decrease prejudice and hostility and help resolve conflict on the individual level. This direction was shown to lead to reduction of intergroup hostility under some conditions[9,40], especially during supervised interactions aimed at building personal connections. Nevertheless, it has had limited effect in competitive, unsupervised conditions[41].

One way that intergroup bias may be maintained is through biased learning mechanisms. Studies have shown that individuals need to learn about the individual traits of the other in order to predict that person's behavior and adapt one's own behavior accordingly[42–44]. For example, by accumulating advisor advice accuracy over multiple experiences, people can learn and make inferences about an individual's honesty[45]. Based on one specific situation, people tend to infer not only an individual's general traits, but also the general traits of all members of that individual's social group[42,46–48]. Previous research indicated that people tend to treat information about ingroup and outgroup members differently. They tend to assume that ingroup and outgroup members have different motivations for the same actions[34,48]. For example, selfish behavior may be interpreted as self-serving when performed by an ingroup member, but as hostile and harm-seeking when performed by an outgroup member[49]. In addition,

group membership may bias one's attention, making an observer more likely to detect cooperative cues from an ingroup member[50]. Finally, people also tend to perceive outgroup members as similar to one other, while perceiving greater variability between ingroup members, in an effect termed outgroup homogeneity[51,52]. Such biases in perception may shape learning and influence how learners update their impressions of ingroup and outgroup members based on experience, such that these impressions are dependent on the learner and the target group affiliation.

This study seeks to examine how social learning processes are shaped by social identity and how they support the formation, maintenance, and facilitation of intergroup biases. We hypothesized three mechanisms by which social learning can be biased: biased prior beliefs, biased learning rates, and biased group-level attribution. First, learners' prior assumptions about the likelihood another person will perform a competitive act may be dependent on that person's group affiliation. Second, learning from competitive and cooperative actions may be group dependent, such that cooperative actions exert a greater influence on learning about ingroups than do competitive actions, and vice versa in the case of learning about outgroup members. Finally, group-level attribution may affect learning about ingroup and outgroup members differently and may operate differently on competitive and cooperative actions. For example, it may be manifested in a higher degree of group-level attribution of competitive actions from one outgroup member to another, making all outgroup members seem more likely to perform competitive acts.

To test these hypotheses, we adapted a sequential social dilemma paradigm called the star-harvest game[42,53], in which five players collect stars and are allowed to sacrifice a move to zap other players and send them to a time-out zone for three turns (Fig. 1A). We employed a computational social learning model to this game to study how trial-by-trial experiences and asymmetric learning processes shape intergroup bias. Participants played the star-harvest game with four other players. Participants were not

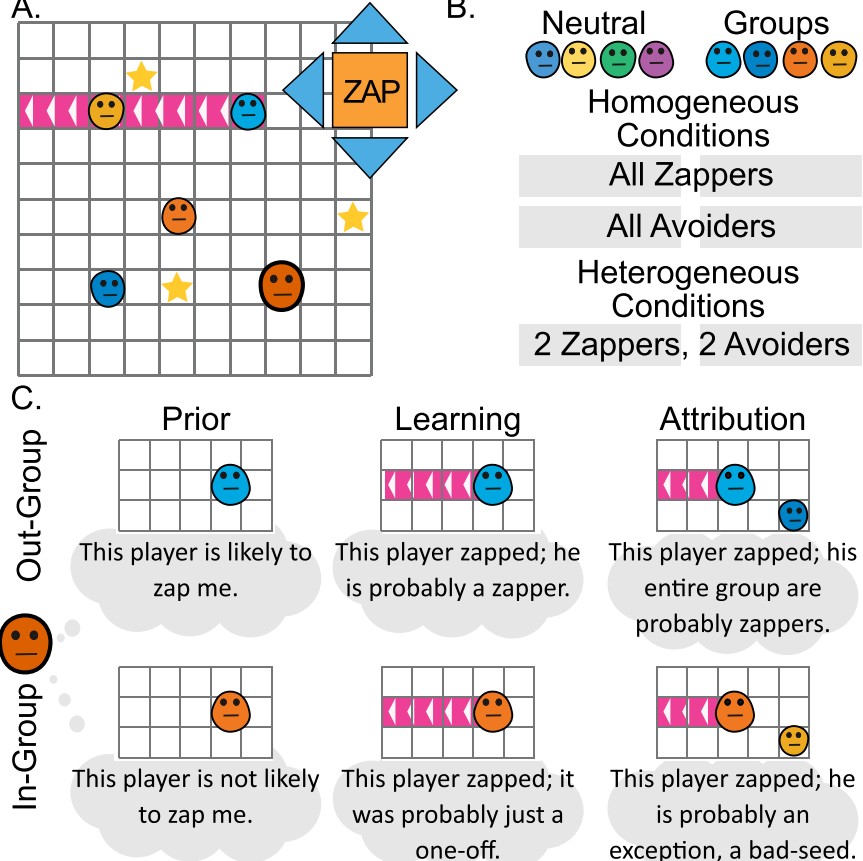

**Fig. 1 | Minimal-group star-harvest task and main hypotheses. A** The game layout consisted of five players—the participant (marked with thick border) and four other bot-players—who moved across a 2D grid and collected stars. On each trial, players could either zap each other (via a ray that sent the affected other player to a time-out zone for three turns) or could avoid zapping and move using the blue arrows. The stars they collected were presented as their score. **B** The experiment included six conditions. In the Groups conditions, participants initially picked a team color for themselves, and then played with two same-color and two different-color bot-players. In the Neutral conditions, participants picked a color for their player and played with four bot-players with different colors. In the homogeneous conditions all bot-player behaved in the same way, either as zappers or zap-avoiders. In the heterogeneous conditions, two bot-players (one from each color team) were zappers and the other two bot-players were zap avoiders. **C** Three hypothesized learning mechanisms can govern learning about the zap behavior of the other and can be affected by the other player's group identity.

informed about the identity of the other players. Those players were in fact bot-players and were programmed to manifest one of two behavioral patterns. All bot-players were programmed to display star-seeking behavior, and differed in the way they react when another player was standing between them and a star. One type of bot-player zapped this player, clearing the path for them to collect the star, a type called 'zappers'. The other type of bot-player did not zap, and instead moved away towards other stars, a type called 'avoiders' (see methods for full details). To test the group identity effect, we used the minimal-group procedure[16,17], in which participants chose the team color of their avatar. They played with two bot-players with the same avatar team color (ingroups) and two bot-players with the other team color (outgroups). The color of the avatars did not affect their behavior, such that ingroup zappers and avoiders behaved in the same way as outgroup zappers and avoiders. Finally, our experimental design included homogenous conditions, in which all bot-players behaved in the same way, and heterogenous conditions, in which two bot-players were zappers and two bot-players were avoiders (Fig. 1B). This experimental design allowed us to examine the preregistered effects of the three learning mechanisms on behavior using a computational learning model (Fig. 1C). First, prior effects were expected to be manifested in a generally greater likelihood that outgroups would be zapped more than ingroups, and markedly so in the all-avoiders condition, in which both ingroups and outgroups never zap. Second, a learning effect was expected to be manifested in differential treatment of zappers and avoiders from the different groups. Finally, an attribution effect was expected to be found in differences in the way zappers and avoiders are treated in homogenous and heterogenous groups. All three effects can be characterized by our computational learning model.

## Methods
### Participants
We recruited participants from the Prolific online platform. Participants were randomly assigned to six experimental conditions that comprised all-zappers, all-avoiders and mixed, and within these to group and neutral conditions, for a total of six conditions (Fig. 1B). The group conditions included two subtypes in which the order according to which players played each turn (ingroup and outgroup) was counterbalanced. We therefore collected twice as many participants in the group conditions than in the neutral condition. We preregistered a sample size of $N = 630$ based on a preliminary pilot study pointing to an effect size of $f2 = 0.02$ for the interaction effect in zap ratings, to achieve 90% power in detecting this effect size, with alpha level of 0.05. We collected data from a total of 680 participants (357 men, 316 women, 2 non-binary, and 5 choosing not to indicate gender), with a mean age of 37.2 (±12.34). (See full details per experimental condition in supplementary Table 1– in the supplementary materials.) No participant was excluded from the main analysis. All participants gave their informed consent and received monetary compensation at a fixed rate of GBP 2.5 for 15 min of participation (resulting in a mean hourly rate of GBP11.5). The study was approved by the research ethics committee of the Faculty of Social Sciences at the University of Haifa, Israel (number 038/18).

### Experimental Task
The experimental task builds on the star-harvest game, which was previously developed to provide a flexible and rich setting for studying different types of social learning mechanisms in a user-friendly manner[42]. The game included five players, represented by colored avatars that move around a 10 × 10 grid (Fig. 1, see example here: http://socialdecisionlab.net/stuff/GridWorldDemoN/). The game is played on a turn-by-turn basis, and the order of players remains constant throughout the game. On each turn players can either move in one of four directions and collect stars that appear on the grid, or they can zap by emitting a pink ray in one of the four directions. Players caught in the ray are sent to a time-out zone visible to the player for three turns. After each round in which all players take a turn, a new star can appear anywhere on the grid with a 0.75 probability, and uncollected stars can disappear. Each player's collected stars appear in their

'score' section on the screen. The participants did not receive any bonus based on the stars they collected, beyond the fixed monetary rate.

Participants were instructed that they are not going to act alone. In the group conditions the task was preceded by a minimal group manipulation[15,54], in which the participant was asked to choose a team-color for his avatar (blue or orange) and was shown the avatar colors of his fellow players, two of which were the same color (but different shade) as the participants, and two were in the color scheme of the other, unchosen, color. In the control condition participants chose one of five different colors for their avatar. See Fig. 1 and the supplemental material for full instructions. Please note that in light of the nature of the study, debriefing about the mild deception was not a requirement by the ethics committee.

The behavior of the bot-players was governed by algorithms implementing zapper and avoider behavior, as used in a previous study[42] (Supplementary Fig. S1). Both bot-player types were programmed to first check whether they are the closest player to a star, and if so to move towards the star, thus concluding their turn. Otherwise, bot-players were programmed to check whether they are in direct competition with another player for a star. This happens when another player is on their way to a star, i.e., closer to the star closest to them. Avoiders were programmed to move away and seek other stars. Zappers were programmed to zap the competing player if they share the same row or column. Zapping is therefore done in the context of competition over a star, and not arbitrarily, i.e., does not occur every time the bot-player *can* zap another player. The two algorithms capture the behavior of star-seeking players with different levels of competitiveness.

### Procedure
Participants were directed to the experimental task website and received instructions about the game play. They were then told that they would play the game with four other players, but were not explicitly informed about the players' identity, and whether they were humans or bots. They then continued to choose their avatar color, and to the task. The task lasted 100 turns; each turn included actions from all five players (unless they were in the time-out zone). The order of players' moves on each turn was kept constant. We collected the location of players and stars on each turn and move, and the actions carried by the players. The game lasted 12 minutes on average.

At the end of the task, we included three items to examine the effects of the group manipulation and the participants' experience in the task. First, we asked them to distribute ten extra stars between the four players with whom they played. All ten stars had to be allocated. Second, we asked participants to rate the intentions behind the behavior of the other players. We asked them to rate harmful intent ('prevent others from gaining stars') and selfish intent ('gain more stars'), as previous research found intergroup effects on such judgments[49].

### Computational learning model
We used a computational learning model to identify the contribution of three mechanisms, prior, learning and group level attribution, to the way participants decide whether or not to zap other players. Our model includes a decision rule by which participants make the decision to zap other players on a trial-by-trial basis, and a learning mechanism which accumulates the behavior of the other players to establish beliefs about their likelihood to zap.

**Decision rule.** On a trial-by-trial basis, we modeled the decision to zap or avoid zapping a target player as dependent on a weighted sum of multiple variables (eq. [1]). These variables included the situational factors of distance from the target (weighted by free parameter $w_{DistTarget}$) and distance from the closest star (parameter $w_{DistStar}$). It was also dependent on zap priors: a zapping bias (parameter $Bias$), indicating the overall inclination of participants to zap, and prior related to the group identity of the target, either ingroup or outgroup (parameters ($Prior_{in}/Prior_{out}$)), indicating the likelihood to zap players based only on their group identity, regardless of their behavior. Finally, the decision was dependent on the learned belief about the target's likelihood to zap (parameter $w_{TargetZap}$ $w_{TargetZap}$). This belief is not immediately available from the data and has

**Fig. 2 | Model simulations of three learning mechanisms.** We simulated the learning model to generate zapping behavior of agents in the heterogeneous conditions which include zapper and avoider bot-players. We examine the pattern of simulated agent's zapping behavior (right) and the underlying learning curves (left) which represent the way beliefs about players' likelihood to zap are learned over time. **A** We disabled learning and attribution effects and set the group identity prior parameters to be either low (–0.9) or high (–0.1). The model predicts high zapping rates when prior is high, and low zapping rates when prior is low, and no learning about players' behavior. Note that situational variables such as distance from target, and the tendency of zappers to be closer to other players, affected simulated zapping behavior. **B** We disabled the attribution effect, fixed the prior effect and varied the learning rates (LR) of zaps to be either low (0.2) or high (0.9). Learning curves illustrate faster learning of zappers behavior (left), and difference in zapping rates between zappers and avoiders (right). **C** We fixed zapping learning rates and priors and varied the zap attribution rate parameter to be either low (0.2) or high (0.5). High attribution rates increased the estimation of the zapping probability of zap avoiders (left) and led to more similar zapping behavior towards zappers and avoiders (right). Lines in learning curves indicate mean zapping estimation variables, shadows indicate 95% confidence intervals of the mean. Boxplots include the median in bold line, interquartile range is represented by the box, minimum and maximum range by the whiskers, and outliers by dots.

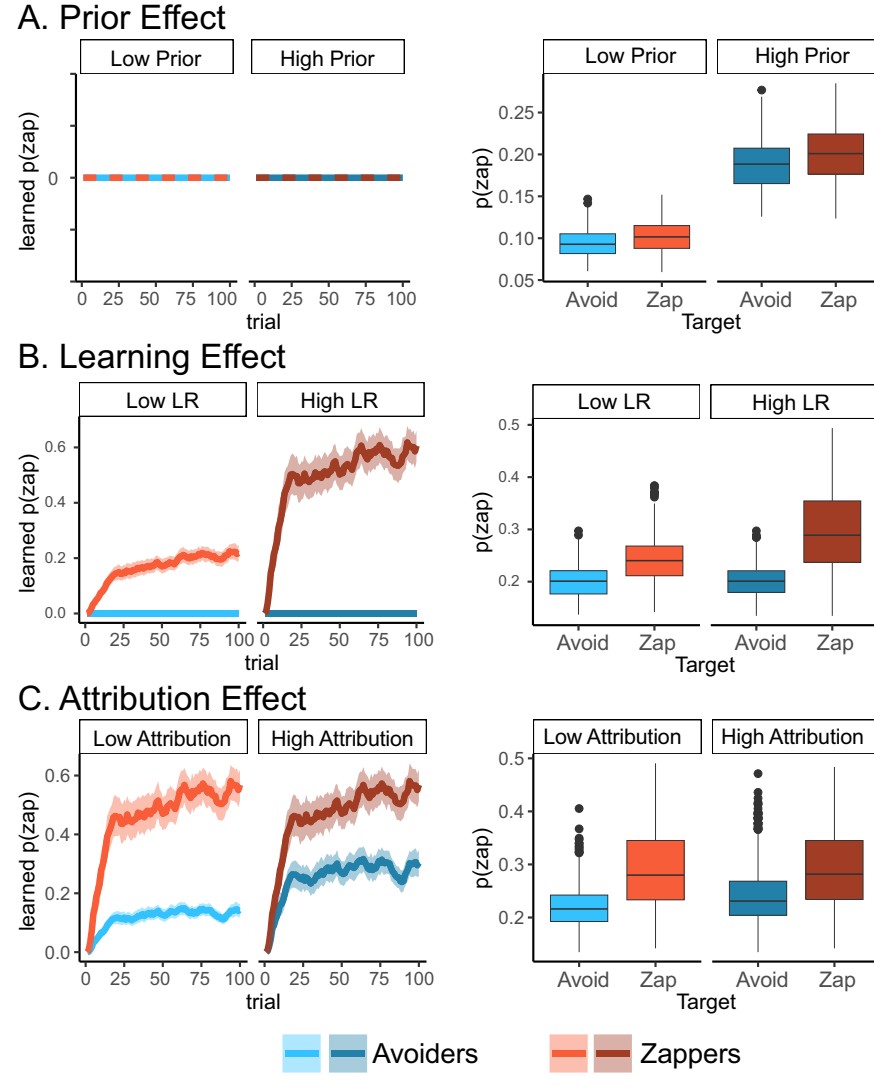

to be inferred using the learning part of the model.

$$p(Zap) = Logit(Bias + Prior_{in}/Prior_{out} + w_{StarDist} \cdot StarDist + w_{TargetDist} \cdot TargetDist + w_{TargetZap} \cdot TargetZap) \quad (1)$$

### Learning mechanism
To estimate the likelihood of each of the four other players to zap, we used a reinforcement learning mechanism that calculates the prediction error, i.e., the difference between the observed behavior (zap = 1, avoid = 0) on each trial and current estimation of player's likelihood to zap, and updates the belief using a learning rate[55] (eq. [2]). We assumed different learning rates for avoidance and zapping behavior for in/outgroup members (four free parameters). The initial value for beliefs about players' likelihood to zap was set to 0, as prior beliefs were captured by the $Prior_{in}/Prior_{out}$ parameters in the decision rule.

$$PlayerZap(t+1) = PlayerZap(t) + \begin{cases} LR_{Zap}^{In/Out} \cdot (1 - PlayerZap(t)) \\ LR_{Avoid}^{In/Out} \cdot (0 - PlayerZap(t)) \end{cases} \quad (2)$$

Group-Attribution was modeled in terms of updating the beliefs not only about the player that just acted, but also about his other group members. We assumed different group-attribution values for avoidance and zapping behavior for in/outgroup members (four free parameters). These free parameters governed the rate of update, similar to learning rates. When attribution rates are close to 0 it means no group level attribution, and when they are identical to the learning rates (LR) it means complete attribution.

$$OthersZap(t+1) = OthersZap(t) + \begin{cases} At_{Zap}^{In/Out} \cdot (1 - OthersZap(t)) \\ At_{Avoid}^{In/Out} \cdot (0 - OthersZap(t)) \end{cases} \quad (3)$$

A number of technical decisions were made in setting up the learning model. First, while it is straightforward to detect zapping behavior, and update the beliefs about the zapper, detecting avoidances is less clear. Here we defined avoidances as moving (instead of zapping) when the player was able to zap another player, i.e., shared a column or row with him, and when the player was not closest to a star, in line with our bot-player algorithm (Supplementary Fig. S1). This means that no learning occurred when the observed player was closest to a star and moved towards it, or when the observed player could not zap any other player.

Another assumption is that our model tries to predict zapping behavior only when the participants could zap another player, i.e., shared a row or column with another player. Our model did not provide predictions for trials when the participants could not zap other players. This also means that these trials did not contribute to the model fitting procedure.

## Model simulations

We simulated the computational learning model to examine how well its predictions of zapping behavior can differentiate between the learning mechanisms. To simulate the model, we used data collected in a pilot study from 189 participants in the six experimental conditions. The data we used included the stars and bot-players' locations on each turn and move, and their behavior; moves, zaps and avoidances. This allowed us to predict how a player governed by our learning and zap-decision model would behave in our experiment, in a variety of situational contexts.

The simulations included three steps, from a simple model with no learning and attribution mechanisms (prior only, P model), to one that includes the learning mechanism (prior+learning, PL model), and a model that includes the attribution mechanism (prior+learning+attribution, PLA). We did not simulate all parameter space, but instead focused on two values for one set of parameters in each simulation to demonstrate the models' flexibility and its ability to capture different behavioral patterns in our experimental design.

Finally, after the model fitting procedure, we simulated the model using the estimated group-level parameters, and the data from the experiment proper, to generate the model zapping predictions and the underlying beliefs about player's likelihood to zap.

## Analysis

To evaluate differences in zapping behavior and in star allocations, we used mixed-effects linear regressions with group-level coefficients (fixed effects) to model population-level effects and individual-level coefficients (random effects) to capture average individual responses[56]. We report type III ANOVA based on the regression, including F values, p values and partial eta square as effect size index. All analyses were conducted using R software (R version 4.2.2). Analysis packages are detailed in the supplementary materials.

The model fitting procedure was conducted using the Hamiltonian Monte Carlo engine STAN via the 'rstan' package[57]. We used hierarchical model fitting, in which group-level parameter averages and standard deviations were fitted and individual-level parameters could also be extracted. We used weakly informative normal priors centered on 0, which were estimated on a linear scale for weight parameters and prior parameters (eq [1]), and a logit scale for priors, learning rate and attribution rates. To improve convergence, we implemented the non-centered version of varying effects using a Cholesky decomposition of the correlation matrix[58] Model comparison was estimated using 'rethinking' R package[59]. We calculated model's WAIC scores and compared them by calculating the difference between WAIC scores (dWAIC) and the standard error of the difference (dSE), and the Aikake weight given to each model when considering their ensemble predictions.

Sample size and analyses were pre-registered, based on a pilot experiment.

Preregistration: https://osf.io/75mdf/?view_only=949067f8651d4878b055a05ef78a7108

## Reporting summary

Further information on research design is available in the Nature Portfolio Reporting Summary linked to this article.

## Results

### Model simulations

First we evaluated how the three learning mechanisms—prior effects, individual learning, and group-level attribution— can be captured by a computational model developed in previous work and how they can produce different learning patterns in our experimental design[42]. Our model includes a decision rule by which participants make the decision to zap other players on a trial-by-trial basis. It also included a learning mechanism, which establishes beliefs about different players' likelihood to zap on a trial-by-trial basis based on prior, individual-level learning and group-level attribution. Full details regarding the model are included in the methods section.

To examine our model and experimental design ability to capture the effects of different learning mechanisms, we simulate the model using data collected in a pilot study. For the model simulations, we discarded the participants' behavior and used only the bot-players' behavior (location and zaps) and the star locations to generate expected zapping behavior under different experimental conditions. In each simulation, we either disabled two learning mechanisms or kept them fixed and varied the model parameters associated with the mechanism of interest. We examined the learning patterns in the heterogeneous conditions, where different contributions from the learning mechanisms were expected to be most pronounced. We retrieved the trial-by-trial beliefs about players' zap probabilities estimated by the model, and the zapping rate towards each player (Fig. 2).

We began examining how prior effects shape beliefs and zap behavior by setting learning and attribution rates to zero. We used two values for priors—low (–0.9) and high (–0.1). The model's estimation of beliefs about players' zap probabilities did not change, regardless of the players' zap behavior or the behavior of their group members (Fig. 2A). The zapping pattern was dependent on the priors' value, with a higher likelihood to zap when prior value was high compared to low.

We then examined how learning rate values shape learning by fixing the prior parameters and attribution-rates to 0. We also kept the avoidance learning rate fixed at 0.25, and zap learning rate to be either low (0.2) or high (0.9). The estimation of beliefs about players' zapping probability was dependent on the players' behavior, increasing only for zappers and not for avoiders. The model predicted faster learning and higher zapping beliefs when learning rates were high. These beliefs were translated to higher zapping rates for zappers compared with avoiders, and higher zapping rates when learning rates were high.

Finally, to evaluate the group-level attribution effects, we kept priors fixed at 0, zapping learning rates fixed at 0.8 and avoidance learning rates fixed at 0.25. We also fixed the avoidance attribution rates at 0, and set the zapping attribution rates to be either low (0.2) or high (0.5). The model predicted that beliefs about players' zapping behavior would be higher for zappers compared with avoiders, but that estimation of avoiders' likelihood to zap would increase over time even though they never zapped, due to the group-level attribution. High attribution rates led to increased similarity in beliefs about players' zapping behavior between the zappers and avoiders. This translated to increased similarity in zapping rates of zappers and avoiders when attribution rates were high.

To summarize, our simulations showed that our learning model, coupled with our experimental design, can capture the three different learning mechanisms and that setting different values to parameters in the model can yield different learning patterns in our experimental task and its conditions.

### Participants' aggregated behavior

We collected data from 680 participants who played the different experimental conditions online in a pre-registered experiment (see Methods). To examine participants' behavior in our task, we first examined the rate at which participants zapped the other players. According to our preregistered analysis, we calculated the number of times each participant zapped each of the bot-players and the number of turns on which said participant could zap each of the bot-players, i.e., share a column or a row on the grid with the bot-players. We divided these measures to calculate the zapping rate (the percentage of zaps out of all zap opportunities). We used this measure as a dependent variable in a mixed-effects linear regression, which included group identity (ingroup/outgroup/neutral), bot-player's behavior (zapper/avoider), and group homogeneity (homogeneous/heterogeneous), and the interactions between these factors.

Our analysis revealed a significant effect of group identity ($F_{(2,1011.76)} = 179.38$, $p < 0.001$, $\eta_{partial} = 0.26$, 95% CI (confidence interval): [0.22, 0.3]) and a significant effect of bot-player behavior ($F_{(1, 1400)} = 77.84$, $p < 0.001$, $\eta_{partial} = 0.052$, 95% CI [0.032, 0.077]) (Fig. 4 and supplementary table 2 in the supplementary materials). We did not observe significant

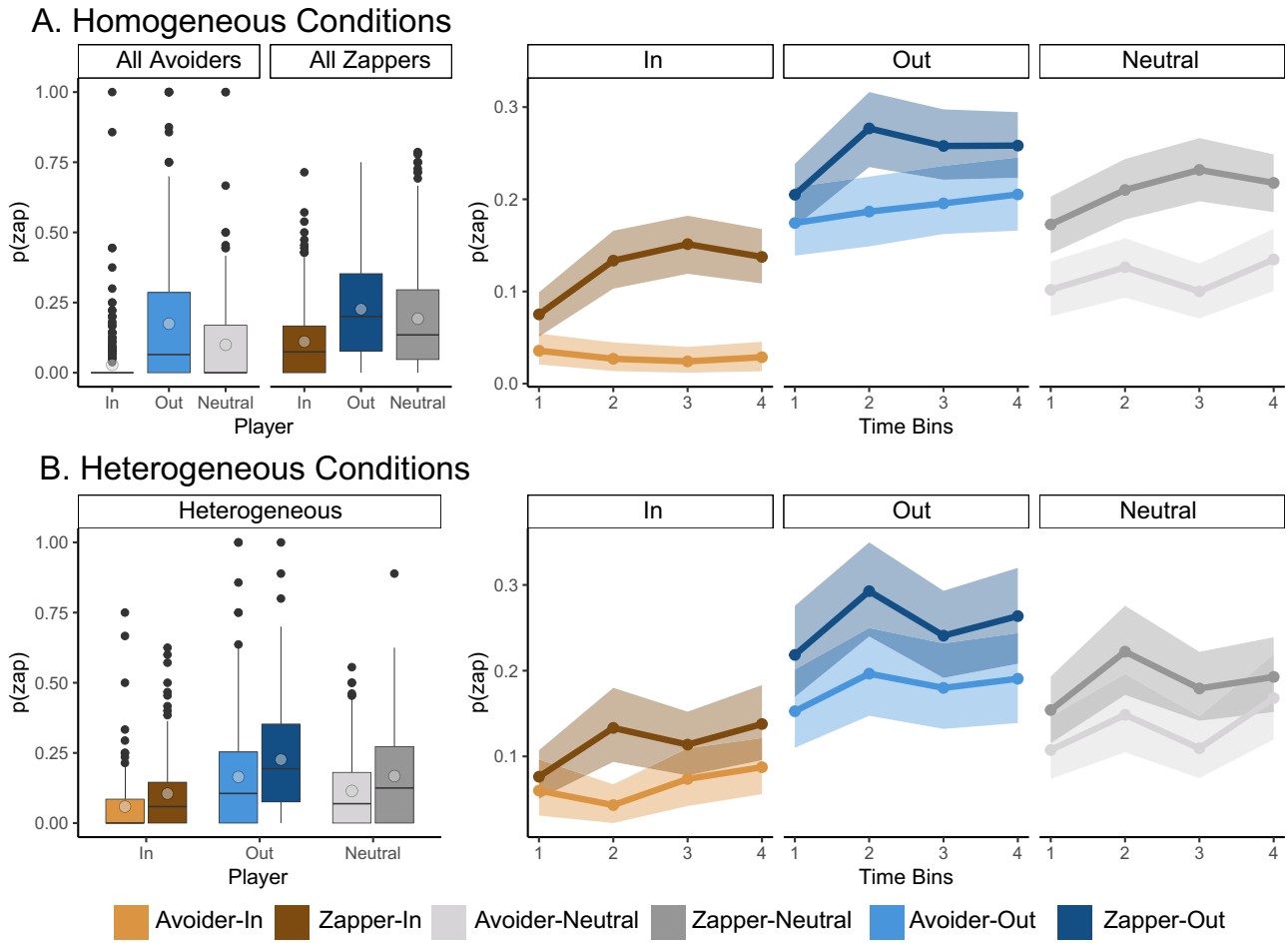

**Fig. 3 | Group-identity and bot-players' behavior effects on participants' zapping behavior experimental outcomes.** Zapping rates throughout the experimental block (left panels) and progression of zapping rates throughout the blocks (right panels) in the homogeneous (**A**) and the heterogeneous (**B**) conditions. Participants zapped outgroup bot-players more than neutral and ingroup bot-players, indicating a consistent intergroup bias. Participants were also more likely to zap bot-players that displayed zapping behavior than zap-avoiders and increased their likelihood to zap zappers over time, indicating learning effect on behavior. In the heterogeneous condition participants showed reduced differentiation between avoiders and zappers in the progression of zapping rates, indicating a group-level attribution effect. Lines in learning curves indicate mean zapping frequencies, shadows indicate 95% confidence intervals of the mean. Boxplots include the mean by light circles, median in bold line, interquartile range is represented by the box, minimum and maximum range by the whiskers, and outliers by black dots.

interaction effects. The behavior effect is indicative of a learning effect, as participants adjusted their zapping behavior according to the observed behavior of the bot-players and were more likely to zap players that displayed zapping behavior. We also observed a group identity effect, as participants were more likely to zap outgroup than ingroup bot-players, with neutral bot-players in the middle, even in cases where no bot-player was zapping (all-avoiders conditions). This indicates a persistent intergroup bias, as zapping behavior was strongly dependent on bot-players' group identity, even when all bot-players behaved in the same manner. We also predicted that due to group-level attribution we would observe a difference in zapping behavior between the heterogeneous and homogeneous conditions. Nevertheless, we did not find a significant interaction effect in the aggregated zap rates measure.

We followed this analysis with an exploratory time-bin analysis, calculating zapping rates in four consecutive 25-trial long time bins, to examine how observation of bot-players' behavior throughout the experiment shaped participants' zapping decisions (Fig. 3 and supplementary tables 3 and 4 in the supplementary materials). We used mixed-effects linear regression, which included group identity (ingroup/outgroup), bot-player's behavior (zapper/avoider), and a continuous time-bin factor (1–4), and the interactions between these factors as independent variables, and zapping frequency as dependent variable. We analyzed the homogenous and heterogeneous conditions independently. In both conditions we found a

significant group effect (Homogeneous): ($F(1, 1798) = 48.5$), $p < 0.001$, $\eta_{partial} = 0.012$, 95% CI [0.01, 0.02], Heterogeneous: ($F(1, 1798) = 29.3$, $p < 0.001$, $\eta_{partial} = 0.014$, 95% CI [0.01, 0.03]), in line with the full-block analysis. In the homogeneous condition we found a significant time-bin effect ($F(1, 1798) = 9.49$, $p = 0.002$, $\eta_{partial} = 0.002$, 95% CI [0.003, 0.006]), as participants' zapping behavior increased throughout the block, and did not observe a significant behavior effect ($F(1, 1798) = 2.94$, $p = 0.08$, $\eta_{partial} = 0.002$, 95% CI [0.00, 0.01]), but found an interaction between time-bin and behavior ($F(2, 1798) = 5.27$, $p = 0.021$, $\eta_{partial} = 0.001$, 95% CI [0.0001, 0.004]), as likelihood to zap avoiders did not increase over time whereas likelihood to zap zappers increased over time. This demonstrates a learning effect on zapping decisions, and this effect is dependent on the players' behavior. In the heterogeneous condition we found a significant time-bin effect ($F(1, 1798) = 6.23$, $p = 0.012$, $\eta_{partial} = 0.003$, 95% CI [0.00, 0.01]), and a significant behavior effect ($F(2, 1798) = 4.6$, $p = 0.03$, $\eta_{partial} = 0.002$, 95% CI [0.00, 0.008]), but did not observe a significant interaction effect ($F(2, 1798) = 0.06$, $p = 0.79$, $\eta_{partial} < 0.0001$, 95% CI [0.00, 0.002]).

We examined several other behavioral measures that can indicate intergroup bias in our experiment. In an exploratory analysis, we examined the likelihood of participants to cross-path with another player, i.e., to choose to move to a location where they share the same row or column with another player. Cross-pathing means that the participant becomes

**Fig. 4 | Additional behavioral manifestations of intergroup bias. A** Path-cross frequency, indicating the frequency of participants deciding to move to a position where another player can zap them. Participants were more likely to cross-path with ingroup than outgroup players, and with avoiders compared with zappers. **B** After the experiment, participants were asked to distribute an extra ten stars between bot-players. They allocated more stars to ingroup bot-players than to outgroup players. In heterogenous conditions they allocated more stars to zap-avoiders, except in the outgroup condition, in which both group members were treated similarly. Boxplots include the mean by light circles, median in bold line, interquartile range is represented by the box, minimum and maximum range by the whiskers, and outliers by black dots.

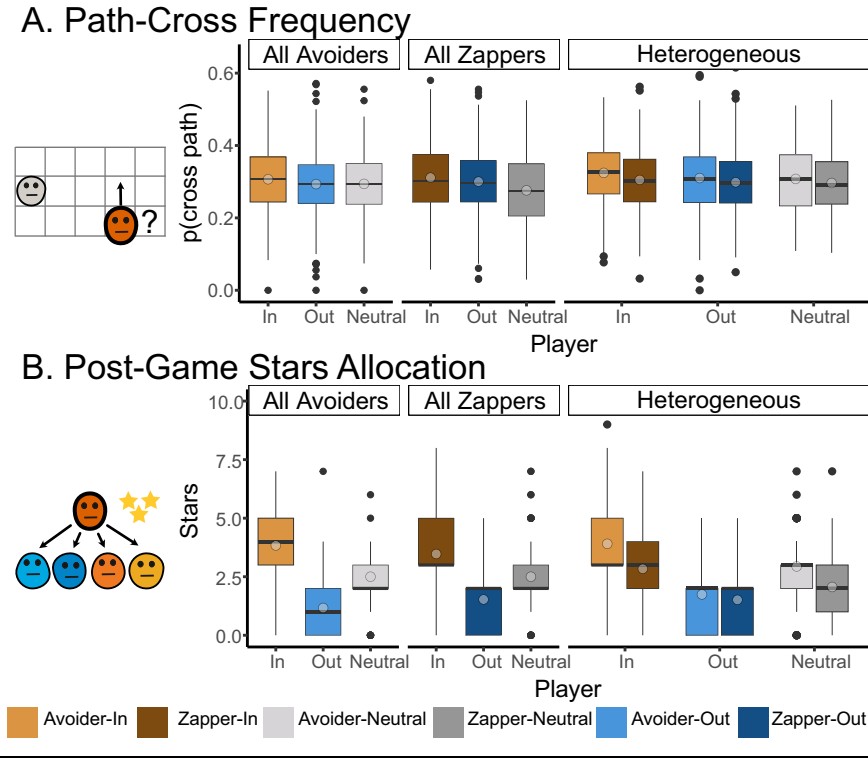

vulnerable as the other player can zap them in the next trial (Fig. 4A). We used this measure as a dependent variable in a mixed-effects linear regression, which included group identity (ingroup/outgroup/neutral), bot-player's behavior (zapper/avoider), and group homogeneity (homogeneous/heterogeneous), and the interactions between these factors as independent variables. Our analysis revealed a significant effect of group identity ($F_{(2, 1012)} = 6.65$, $p = 0.0013$, $\eta_{partial} = 0.013$, 95% CI [0.002, 0.03]) and a significant effect of bot-player behavior ($F_{(1, 1983.5)} = 4.21$, $p = 0.04$, $\eta_{partial} = 0.002$, 95% CI [0.00, 0.01]) (Supplementary Table 5 in the Supplementary Materials). Participants were more likely to cross-path with ingroups, and avoid outgroups, indicating an intergroup bias in their estimation of the risk posed by ingroup and outgroup players. In addition, participants were sensitive to the behavior of the players, and were more likely to avoid zappers and to cross-path with zap-avoiders.

After the experiment was over, participants were asked to perform two additional pre-registered tasks. First, we asked participants to rate how much they thought each player's actions were motivated by their desire to gain stars, i.e., selfish intention. We ran a mixed-effects linear regression, with selfish intentions ratings as dependent variables, and bot-player group-identity (ingroup/outgroup/neutral), bot-player behavior (zapper/avoider), and condition (homogeneous/heterogeneous) and their interactions as dependent variables. Results show that participants tended to rate ingroups as acting from selfish motive more than outgroups ($F_{(2,1012)} = 17.85$, $p < 0.001$, $\eta_{partial} = 0.03$, 95% CI [0.01, 0.06]), and that zappers were rated as more selfish than avoiders ($F_{(1,1105.9)} = 14.09$, $p < 0.001$, $\eta_{partial} = 0.01$, 95% CI [0.00, 0.03]). Participants tended to rate selfish intentions higher in the heterogeneous conditions than in the homogeneous conditions, indicating an attribution effect ($F_{(1,674)} = 6.84$, $p = 0.01$, $\eta_{partial} = 0.01$, 95% CI [0.00, 0.03]) (see for further details in Supplementary Fig. S2 and supplementary Table 8).

Participants were also required to rate how much they thought different players acted based on an intention to prevent others from gaining stars, i.e., harm intention. We ran a mixed-effects linear regression, with harm intentions ratings as dependent variables, and bot-player group-identity (ingroup/outgroup/neutral), bot-player behavior (zapper/avoider), and condition (homogeneous/heterogeneous) and their interactions as dependent variables. Results show that participants tended to rate zappers as

acting more from harmful motive than avoiders ($F_{(1,1196.8)} = 341$, $p < 0.001$, $\eta_{partial} = 0.22$, 95% CI [0.18, 0.26]). Participants also tended to rate harmful intentions higher in the heterogeneous conditions than in the homogeneous conditions, indicating an attribution effect ($F_{(1,674)} = 55.12$, $p < 0.001$, $\eta_{partial} = 0.08$, 95% CI [0.04, 0.11]) (more details in Supplementary Fig. S3 and supplementary table 8 in the supplementary materials).

In addition, participants were also asked to distribute ten extra stars among the four players they had played with. We ran two separate analyses for the homogeneous and heterogeneous conditions, with stars-allocated as the dependent variable and bot-player group-identity (ingroup/outgroup/neutral), bot-player behavior (zapper/avoider), and their interaction as dependent variables (Fig. 4B and Supplementary Tables 6 and 7 in the supplementary materials). In the homogenous conditions of all-zappers and all-avoiders, participants distributed the stars equally in the neutral conditions and favored ingroup over outgroup bot-players in the grouped conditions ($F_{(1, 1798)} = 475.9$, $p < 0.001$, $\eta_{partial} = 0.34$, 95% CI [0.26, 0.38]), in line with a prior effect (Fig. 4B). We also found a significant interaction effect ($F_{(1, 1798)} = 11.9$, $p < 0.001$, $\eta_{partial} = 0.013$, 95% CI [0.004, 0.024]), as the gap in star allocations between ingroup and outgroup bot-players narrowed in the all-zappers condition compared with the all-avoiders condition, in line with the learning effect and increased zapping rates. When examining the heterogenous conditions, we found a significant effect of group identity ($F_{(2, 910)} = 82.16$, $p < 0.001$, $\eta_{partial} = 0.15$, 95% CI [0.11, 0.19]), such that participants allocated more stars to ingroup bot-players than to neutral and outgroup players. We also found a significant behavior effect ($F_{(1,910)} = 43.85$, $p < 0.001$, $\eta_{partial} = 0.045$, 95% CI [0.00423, 0.075]). In the heterogeneous conditions when participants played with both zappers and avoiders, they allocated more stars to avoiders than to zappers. We also observed a significant group-identity by behavior interaction effect ($F_{(2, 910)} = 5.24$, $p = 0.005$, $\eta_{partial} = 0.011$, 95% CI [0.001, 0.02]), indicating that participants treated zappers and avoiders differently when these players were members of an ingroup or neutral (Fig. 4B).

Overall, our aggregated behavioral results strongly support the notion that participants learn from experience and tend to zap more and allocate fewer post-task stars to zappers than avoiders. In addition, our results strongly indicate a persistent intergroup bias, according to which participants favored ingroups and were hostile to outgroup members compared

with controls, both in their zapping behavior and in their path-crossing and post-task star allocation. Finally, an attribution effect was observed in the time-bin analysis of zapping behavior, and in the post-experiment star allocation, in which outgroup players were treated similarly regardless of their behavior and received fewer stars than ingroup players.

### Estimated learning model parameters

To reveal the learning mechanisms underlying the persistence of intergroup bias, we fitted the learning model described earlier to the participants' data. We fitted the model only to the mixed-grouped condition, in which participants encountered both zapper and avoider bot-players and both ingroup and outgroup players in the same condition. This allowed meaningful estimation of model parameters for ingroup and outgroup players and for zap and avoidance behaviors. We further excluded participants that zapped

### Table 1 | WAIC fitting scores and model comparisons

| Model | WAIC | SE | dWAIC | dSE | pWAIC | weight |
|-------|------|----|-------|-----|-------|--------|
| PLA | 4206.67 | 86.98 | 0 | NA | 193.92 | 0.92 |
| PL | 4211.63 | 87.13 | 4.95 | 5.66 | 178.47 | 0.08 |
| P | 4251.06 | 87.15 | 44.39 | 12.44 | 155.24 | ≪0.01 |

WAIC scores of each model, sorted from small (better) to large (worse), SE is the standard error of each WAIC, dWAIC is the difference between each model's score and the best model's score, dSE is the standard error of this difference, pWAIC is the measure of effective number of parameters, capturing model's flexibility, and weight is the Aikake weight given to each model in the prediction of participants' behavior[58].

less than once during the experiment from the model fitting procedure, as estimation may be unreliable for such sparse behavior. Hence, we excluded 12 participants and fitted the model to the remaining 132 participants (see Method for full model fitting procedure).

We fitted three different models: The first included only the prior mechanism (P model, eq [1]), the second included prior and learning mechanisms (PL model, eq [1] +eq [2]), and the third included prior, learning and group-level attribution mechanisms (PLA model, eq [1] +eq [2] +eq [3]). Our model fitting and model comparison procedure yielded the lowest WAIC fitting score to the PLA model, indicating that it best described the participants' decisions while accounting for its increased complexity and number of parameters (Table 1). Support for the PLA model was already indicated in the aggregated behavioral results, which demonstrated prior, learning and group-level attribution effects.

Our PLA model fitting procedure resulted in estimation of the posterior of group-level parameters, allowing us to compare the learning mechanisms in learning about the zapping behavior of ingroup and outgroup players (Fig. 5A, Table 2). For the decision rule (eq. [1]), we found a support for a positive effect of distance from stars ($w_{DistStars}$ mean: 0.26, 89% high density interval (HDI): [-0.04, 0.58]), as participants were more likely to zap while stars far from them. We found a strong support for a negative distance from target-player stars ($w_{DistTarget}$ mean: −1.84, 89% HDI: [−2.23, −1.46]), as participants were more likely to zap target-players that were close to them. Importantly, we found a strong support for the effect of the target player's likelihood to za on participants' zapping decisions ($w_{TargetZap}$ mean: 0.89, 89% HDI: [0.53, 1.22]). Finally, an overall negative zap-bias was

## A. Estimated Model Parameters

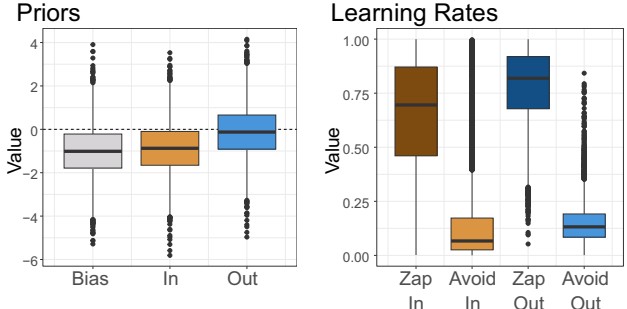
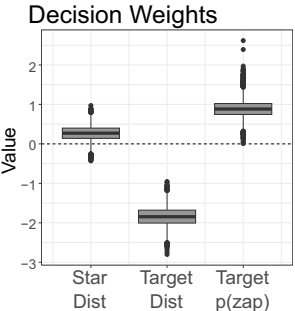

## B. Estimated Learning Pattern

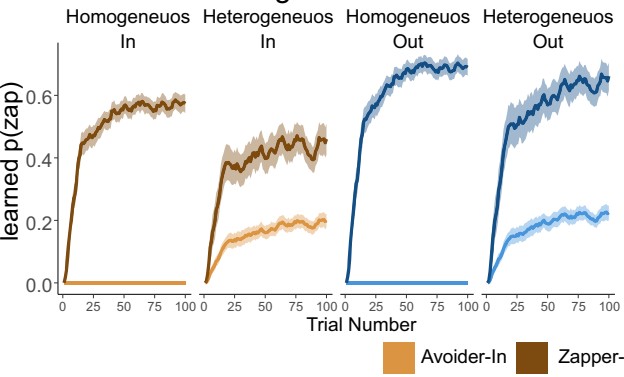

## C. Estimated Zap Pattern

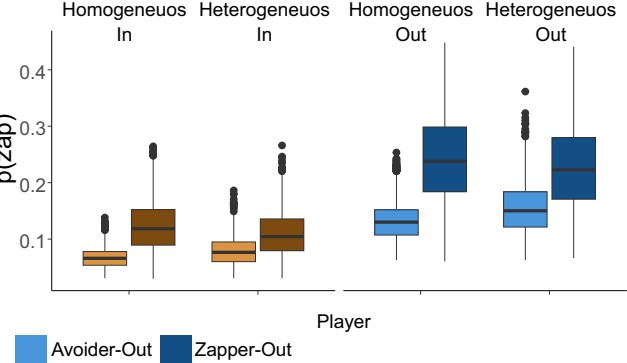

**Fig. 5 | Social learning model results.** We estimated group level parameters of the PLA social learning model. **A** Posterior distribution of the model group-level parameter. 1. Priors include the general bias parameter, and ingroup and outgroup priors. Out-group prior to zap were higher than in-group prior. 2. Learning rates were higher for zaps compared with avoidances and were highest for outgroup zaps. 3. Attribution rates for outgroup avoidance were lower than all other attribution rates. 4. Situational parameters indicate that likelihood to zap increased when stars were far, decreased when the target player was far, and increased when the estimation of the target player's likelihood to zap was high. **B** Model estimations of the

internal learning process about bot-players' likelihood to zap in the different experimental conditions. Higher learning rates for outgroup zaps, and lower attribution rates for outgroup avoidances contributed to higher estimation of zap probability both for outgroup zappers and avoiders. **C** Model predicted zapping behavior shows a group and behavior effect, like the observed behavioral pattern. Lines in learning curves indicate mean zapping estimation variables, shadows indicate 95% confidence intervals. Boxplots include the median in bold line, interquartile range is represented by the box, minimum and maximum range by the whiskers, and outliers by dots.

**Table 2 | Parameter estimations of the computational learning model**

| Variable | Mean | Median | sd | rhat | 89% HDI |
|---|---|---|---|---|---|
| Prior_In | −0.87 | −0.87 | 1.17 | 1.0005 | [−2.76, 0.98] |
| Prior_Out | −0.12 | −0.13 | 1.17 | 1.0005 | [−1.99, 1.76] |
| LR_Zap_In | 0.71 | 0.70 | 0.84 | 1.0016 | [0.27, 1.00] |
| LR_Zap_Out | 0.84 | 0.82 | 0.78 | 1.0009 | [0.55, 1.00] |
| LR_Avoid_In | 0.08 | 0.07 | 0.85 | 1.0020 | [0.00, 0.42] |
| LR_Avoid_Out | 0.13 | 0.13 | 0.69 | 1.0000 | [0.02, 0.27] |
| Bias | −1.01 | −1.01 | 1.17 | 1.0005 | [−2.86, 0.90] |
| w_dist_star | 0.03 | 0.03 | 0.02 | 1.0001 | [−0.04, 0.58] |
| w_dist_Target | −0.18 | −0.18 | 0.02 | 0.9998 | [−2.23, −1.46] |
| w_zap_Target | 0.89 | 0.88 | 0.23 | 1.0007 | [0.53, 1.22] |
| At_Zap_In | 0.19 | 0.19 | 0.83 | 1.0004 | [0.00, 0.60] |
| At_Zap_Out | 0.22 | 0.22 | 0.75 | 1.0009 | [0.01, 0.48] |
| At_Avoid_In | 0.14 | 0.13 | 0.80 | 1.0008 | [0.00, 0.44] |
| At_Avoid_Out | 0.02 | 0.03 | 0.73 | 1.0004 | [0.00, 0.06] |

Mean, median and standard error (sd) of the posterior distribution of group level parameters is presented, along with the 89% HDI. Rhat represents the convergence of the mcmcm chain, with values close to 1 representing convergence.

observed (*bias* mean: –1.01. 89%HDI: [–2.86, 0.90]), indicating that participants tended to avoid zapping overall.

We then examined group-identity dependent differences in parameter values. Group identity prior parameters were different from each other, as ingroup prior was lower than outgroup prior ($Prior_{out} - Prior_{in}$ mean: 0.75, 89% HDI: [0.55, 0.96]). Learning rates were overall higher for zaps than for avoidances within each group ($LR_{zap}^{in} - LR_{avoid}^{in}$ mean: 0.49, 89% HDI: [0.03, 0.99], $LR_{zap}^{out} - LR_{avoid}^{out}$ mean: 0.63, 89% HDI: [0.35, 0.92]), in line with previous works[42]. Learning rates for zaps tended to be higher for outgroup compared with ingroups ($LR_{zap}^{out} - LR_{zap}^{in}$ mean: 0.13, 89% HDI: [-0.27, 0.60]), and learning rates for avoidances were similar between groups ($LR_{avoide}^{out} - LR_{avoide}^{in}$ mean: 0, 89% HDI: [-0.34, 0.31]). Finally, attribution rates for zaps and avoids were similar for ingroups ($At_{zap}^{in} - At_{avoid}^{in}$ mean: 0.06, 89% HDI: [-0.41, 0.67]), but were different for outgroups ($At_{zap}^{out} - At_{avoid}^{out}$ mean: 0.22, 89% HDI: [-0.04, 0.49]), and attribution for avoidances was higher for ingroups compared with outgroups ($At_{avoid}^{in} - At_{avoid}^{out}$ mean: 0.16, 89% HDI: [-0.06, 0.44]), as attribution rates for outgroup avoidances were very low. These results indicate three group-identity asymmetries in learning parameters. First, priors for zaps were higher for outgroup compared with ingroup target players. Second, learning rates for zaps were higher when observing outgroup bot-players compared with ingroup bot-players. Third, group-level attribution of avoidances was lower when observing outgroup bot-players than ingroup bot-players.

To examine how these parameter values shaped learning about other players' zap behavior, we used the mean parameter estimations to simulate the model in all experimental conditions (Fig. 5). We found that our model was able to capture the pattern of participants' zapping frequency. We then inspected the underlying learning curves about the likelihood of the bot-players' likelihood to zap. We found that outgroup's zapping behavior was learned faster, and reached higher plateau, in line with the higher learning rates for outgroups' zaps. We also found that for outgroups, zappers' likelihood to zap did not decrease in the mixed condition, while avoiders' likelihood to zap increased, in line with the asymmetric attribution rates for outgroups. Overall, these results show how asymmetric evidence accumulation and priors make outgroup players appear more competitive, and make intergroup bias persist even when the behavior of players is readily available.

## Discussion

The aim of this study was to examine the way social identity shapes learning about others and the persistence of intergroup bias. We tested the contribution of three learning mechanisms: (1) biased prior effects for ingroups and outgroups; (2) biased learning rates for positive (avoidance) versus negative (zapping) behavior for an ingroup or an outgroup member; (3) biased attribution of behavior to another member. By using a multiplayer star-harvest game and the minimal group procedure, we were able to examine how these learning mechanisms govern trial-by-trial updates of beliefs about the zapping behavior of other players. Participants exhibited intergroup bias in their zapping behavior in that they zapped ingroups less outgroups, even when both behaved in the same way, and were less likely to cross-path with outgroups. In addition, participants showed a learning effect throughout the experiment, and were more likely to zap players that were zappers themselves. Finally, group-level attribution was observed in the similarity of learning patterns between avoiders and zappers' learning patterns and in the post-task star allocation in the heterogeneous condition. We used a computational learning model to uncover the way learning mechanisms underlie the persistence of intergroup bias. We found that participants held higher prior for zapping outgroup players than ingroup players. We also found that the learning rates used to update the participants' beliefs about the players' likelihood to zap were higher for outgroup zaps than for ingroup zaps, and that group-level attribution for outgroup avoidances was lower than for ingroup avoidances. The combined effects of these mechanisms make asymmetry in group-identity priors hard to overcome, as competitive behaviors have more impact on beliefs about outgroups' behavior than cooperative behaviors.

We found a strong prior effect in all our measures, indicating that allocating participants to arbitrary groups is enough to create biased behavior in favor of the ingroup and against the outgroup[19]. This finding is in line with the literature about stereotypes, and especially the stereotype content model in which social structure features, namely competition and status, shape the content of stereotypes[60]. In our study, participants were presented with a competitive context—a video game in which multiple players try to collect stars—and competitive behavior was shown to be highly salient and persistent[42]. In such competitive settings, a stereotype content model predicts a low warmth stereotype, in line with prior beliefs of high zap probability for outgroups. Setting the expectation in such a way is therefore expected to have a long-lasting effect on behavior throughout the task. Furthermore, Reggev et al. (2021)[61] demonstrated that expectation-consistent information is associated with intrinsic value such as food or money, regardless of the source of that social expectation. This also may explain how easily we develop prior beliefs and how hard it is to reduce them.

We also found that participants learned from their experience and observation of other players, updated their beliefs about other players' zapping behavior, and adjusted their behavior accordingly. Computational models of stereotype changes suggest that such observations can be used to re-categorize people into different subgroups, for example categorizing players according to their behavior instead of their color[62]. Theoretical accounts suggest that such processes may, under some circumstances, underlie the impact of interactions between people from different groups on stereotype reduction[39]. However, we observed a moderate change in attitude toward ingroups and outgroups during our task, as zapping behavior towards ingroups and outgroups did never converge. We found that learning was group dependent, and that outgroups competitive actions were more readily learned than ingroups. This effect was demonstrated in a number of studies, showing that people tend to update their beliefs about others in a way that is consistent with their stereotype, and to assign higher weight to negative actions coming from outgroups compared with ingroups[8,15,61,63]. Increased learning rates about outgroups negative behavior can therefore lead to persistent intergroup bias by making negative stereotypes difficult to overcome.

Finally, we found that participants tended to attribute negative behaviors, but not positive behaviors, of one outgroup member to all other outgroup members. For in-group, however, attribution was symmetrical. This was observed both in the model parameters and in the time-bin analysis and post-task star allocation pattern. The symmetrical attribution for in-groups is in line with recent works showing greater learning and more flexible impressions formation about ingroup members[64]. However, our findings suggest a more complex effect of outgroup homogeneity on intergroup bias than initially predicted, demonstrated by our model's group-level attribution rates. Outgroup homogeneity builds on perceptual bias, where outgroups are less distinguishable from one another. This may be related to differential neural processing of same-race and other-race faces[65], which can later lead to differences in social categorization. In another study, racial minorities were perceived as more salient than majorities, indicating a lower sense of variability among other-race faces[66]. In the current study, participants tended to attribute negative behavior from one outgroup member to another, but not positive behavior. This indicates that in the context of social learning, group-level attribution is not symmetrical but is affected by stereotypes and the content of the actions. People are more ready to view all outgroup members as similarly *competitive*, but not similarly *cooperative*. This effect may attenuate the effect of experience on stereotype change by hampering learners' ability to properly track individual behavior, as it is concurrently confounded by evidence from other individuals. Interestingly, we found symmetrical group-level attribution when learning about ingroup members' behavior, which is more inline with homogeneity assumption. This symmetrical attribution led to reduced belief about the zapping behavior of ingroup zappers, as the avoidances of the other group member was also attributed to them. Homogeneity assumptions may therefore serve different outcomes in intergroup learning processes. Together with the prior effect and learning rates, group level attribution contributes to the persistence of intergroup bias.

## Limitation

Several limitations should be considered when interpreting our results. First, we used bot-players to manipulate the behavior the participants were exposed to in the different experimental conditions, which may cast some questions regarding the extent of the ecological validity of this work. Our aim in this work was not to study the formation of group dynamics, such as intergroup bias, but to study the learning mechanisms underlying this well-established phenomenon. This required controlling the behavior of ingroup and outgroup players, that do not display intergroup bias in their behavior and zap/avoid zapping all other players in a similar way, which we achieved by using bot-players. We assumed that in a live interaction setting, participants would have displayed intergroup bias in their zapping behavior, making ingroups and outgroups behave very differently and thus confounding our ability to study the participants learning process. Therefore, our findings regarding learning biases may unfold in a more complex manner in real interactive situations, where in addition to learning biases players actively display discriminatory behavior. We suggest that even in situations where such discrimination may be controlled for, such as in working environments, biased learning may still impact behavior.

It is also important to note that participants were not explicitly told that they were playing with algorithmic players, and it is very likely that they sustained some level of belief that the other players might be other human participants. As interaction with anonymous humans and anonymous bots in online multiplayer games and environments become ubiquitous, and online participants seem to assume that they might be interacting with other participants[67–69]. In previous works, we found that participants playing multiplayer games online were not good at differentiating between human and algorithmic players[70], to the extent that participants were willing to incur monetary loss to gain influence on players whose identity is unknown[71]. In addition, the behavior of the algorithmic agents was designed to be similar to human players, as they prioritized star collection over zapping, and zapped during competition over stars and not arbitrarily[42]. Our behavioral results are that participants differentiated between ingroup and

outgroup members above and beyond the observed behavior of these players. This effect replicates well documented intergroup bias in a variety of experiments and real life situations[9,15,72,73]. We therefore suggest that the use of bot-players was not detrimental to the formation of group-identity and intergroup bias.

Another limitation is that in the current study we used the minimal group procedure, allocating players to groups by a random color. Using more salient group identity, such as political party, religion, or race, might have elicited larger group effects. Nevertheless, our results show the persistence of intergroup bias even under these minimal conditions of minimal group procedure and bot-players, indicating the strength of the design. Future studies may benefit from such manipulations as well as from adding other group facilitating mechanisms, such as group-level incentives.

To conclude, by using a multiplayer star-harvest game with minimal group manipulation we were able to detect persistent intergroup bias, even when participants experienced and observed similar behavior among ingroups and outgroups. Our experimental design allowed trial-by-trial analysis to reveal the computational learning mechanisms underlying this effect, beyond the average aggregated results. Our modeling approach revealed that a combination of learning mechanisms made it difficult to overcome prior stereotypic beliefs about outgroups. This was high learning rates for competitive behavior and low group-level attribution for cooperative behavior for outgroups. Our results suggest a cognitive mechanistic basis that underlies a broad social phenomenon and can be used to expand theoretical approaches to intergroup biases and conflict.

## Data availability

The data that support the findings of this study are openly available in: https://osf.io/nx2hv/.

## Code availability

Scripts: https://osf.io/nx2hv/.

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

## Acknowledgements

U.H. was supported by the Israel Science Foundation (1532/20). O.N. was supported by the Azrieli Foundation. The funders had no role in study design, data collection and analysis, the decision to publish, or the preparation of the manuscript.

## Author contributions

O.N. and U.H. designed and conducted the research, analyzed the data, wrote the original draft, reviewed and edited the paper. U.H was responsible for funding and supervision.

## Competing interests

The authors declare no competing interests.
