## [Peer Review File · Communications Psychology]

7th Aug 23

Dear Dr Nafcha,

Thank you for your patience during the peer-review process. Your manuscript titled "Asymmetric cognitive learning mechanisms underlying the persistence of intergroup bias" has now been seen by 2 reviewers, whose comments are appended below. You will see that they find your work of some potential interest. However, they have raised quite substantial concerns that must be addressed. In light of these comments, we cannot accept the manuscript for publication, but would be interested in considering a revised version that fully addresses these serious concerns.

We hope you will find the Reviewers' comments useful as you decide how to proceed. Should additional work allow you to address these criticisms, we would be happy to look at a substantially revised manuscript. If you choose to take up this option, please highlight all changes in the manuscript text file, and provide a detailed point-by-point reply to the reviewers.

Editorially, we consider three issues key to address.

First, the model you favour as the best explanation of the data does not outperform the other models, and it exceeds these in complexity. The reviewers' feedback confirms this editorial concern, however, the referees point to additional analyses that may provide more persuasive evidence in favour of one of the models. We note that unless these revisions highlight one model as clearly a better fit to the data, your manuscript must entirely reflect this ambiguity and parameters that differ between models (but are not shown to decisively improve fit) must not be interpreted at length.

Second and relatedly, Reviewer #2 lists a number of concerns regarding the analysis, and in particular the decision rule, which we ask you to address.

Finally, Reviewer #1 highlights that it is unclear whether human participants understood they were playing bots, or if they assumed the other players were humans. Neither the current manuscript nor reference 42 are clear on the instructions or debrief, although the link you provide leads to a demo that mentions other "players". This raises two issues. First, please provide further data (e.g. from a debrief) that supports inferences as to how participants perceived their opponents and, if applicable, whether this mattered to participant behaviour. This may require further data collection. Second, please revise the manuscript to ensure that all aspects of experiment and analysis are entirely clear from the description without reference to secondary documents. We impose no upper limit on the length of the Methods section.

If the revision process takes significantly longer than three months, we will be happy to reconsider your paper at a later date, provided it still presents a significant contribution to the literature at that stage.

Please use the following link to submit your revised manuscript, point-by-point response to the Reviewers' comments with a list of your changes to the manuscript text (which should be in a separate document to any cover letter) and any completed checklist:

[link redacted]

Please do not hesitate to contact me if you have any questions or would like to discuss the required revisions further. Thank you for the opportunity to review your work.

Best regards,

Marike

Marike Schiffer, PhD
Chief Editor
Communications Psychology

EDITORIAL POLICIES AND FORMATTING

Editorial Policy: [Policy requirements](https://www.nature.com/documents/nr-editorial-policy-checklist.pdf) (Download the link to your computer as a PDF.)

Furthermore, please align your manuscript with our format requirements, which are summarized on the following checklist:

[Communications Psychology formatting checklist](https://www.nature.com/documents/commspsychol-style-formatting-checklist-article-rr.pdf)

and also in our style and formatting guide [Communications Psychology formatting guide](https://www.nature.com/documents/commspsychol-style-formatting-guide-accept.pdf) .

* TRANSPARENT PEER REVIEW: Communications Psychology uses a transparent peer review system. This means that we publish the editorial decision letters including Reviewers' comments to the

authors and the author rebuttal letters online as a supplementary peer review file. However, on author request, confidential information and data can be removed from the published reviewer reports and rebuttal letters prior to publication. If your manuscript has been previously reviewed at another journal, those Reviewers' comments would not form part of the published peer review file.

* **CODE AVAILABILITY:** All Communications Psychology manuscripts must include a section titled "Code Availability" at the end of the methods section. In the event of publication, we require that the custom analysis code supporting your conclusions is made available in a publicly accessible repository; please choose a repository that provides a DOI for the code; the link to the repository and the DOI must be included in the Code Availability statement. Publication as Supplementary Information will not suffice. We ask you to prepare and upload code at this stage, to avoid delays later on in the process.

* **DATA AVAILABILITY:**

All Communications Psychology research manuscripts must include a section titled "Data Availability" at the end of the Methods section or main text (if no Methods). More information on this policy, is available at <http://www.nature.com/authors/policies/data/data-availability-statements-data-citations.pdf>.

At a minimum the Data availability statement must explain how the data can be obtained and whether there are any restrictions on data sharing. Communications Psychology strongly endorses open sharing of data. If you do make your data openly available, please include in the statement:

We recommend submitting the data to discipline-specific, community-recognized repositories, where possible and a list of recommended repositories is provided at <http://www.nature.com/sdata/policies/repositories>.

If a community resource is unavailable, data can be submitted to generalist repositories such as [figshare](https://figshare.com/) or [Dryad Digital Repository](http://datadryad.org/). Please provide a unique identifier for the data (for example a DOI or a permanent URL) in the data availability statement, if possible. If the repository does not provide identifiers, we encourage authors to supply the search terms that will return the data. For data that have been obtained from publicly available sources, please provide a URL and the specific data product name in the data availability statement. Data with a DOI should be further cited in the methods reference section.

<http://www.nature.com/authors/policies/availability.html>.

REVIEWER EXPERTISE:

Reviewer #1 ingroup/outgroup behaviour, computational modelling

Reviewer #2 ingroup/outgroup behaviour, computational modelling

Reviewer #1 (Remarks to the Author):

Nafcha and Hertz present a behavioral investigation of intergroup bias. In a game paradigm, participants pursued a goal while having potentially hostile interactions (“zapping”) with other (bot) players. Experimental conditions manipulated the group identities of these players (minimal groups or neutral) and their hostile behavior (always zap or never zap). Direct analysis of player behavior showed that participants learned from the bots’ behavior, and displayed ingroup bias in terms of how often the players zapped, and how they allocated rewards after the game. Computational modeling investigated three different learning mechanisms to explain this bias: priors, learning rates, and differentiated attributions to other group members. The results showed clear evidence for an effect of priors, but, the results were less conclusive regarding the inclusion of learning rate or attribution effects.

This paper has a number of strengths. It investigates a clear, computationally-specific set of hypotheses in a well-controlled, well-powered, preregistered experiment. The paper is generally clearly written, and investigates a question of social importance. Below are a number of issues which I think it could benefit from addressing further:

1. My main concern with respect to this work is its external/ecological validity. The paper is somewhat unclear about what the players believed regarding the identities of the other players – did they know they were bots, or did they believe they were real people? Even if they were told the bots were real, or if it was left ambiguous, there are a number of clues that might have led participants to believe they were bots. For example, each individual bot played with a completely consistent strategy of zapping or not, which is quite unlike how humans behave in most contexts. If the participants realized/knew the bots were not real people, this calls into question some aspects of the interpretation of their behavior. Specifically, it would make it difficult to know whether the results reflect people’s genuine intergroup psychological processes, or the application of a social script specific to competitive games with nonhuman players. Videogaming against computer bot players is now ubiquitous in society, and players are well known to engage in a wide variety of non-naturalistic behaviors to exploit or otherwise manipulate these bots. Moreover, if the participants realized they were playing against bots, they may have attributed the bots behavior to the intentions of the experimenter, raising the risk of compliant demand effects. This limitation needs to be discussed at greater length in the paper, and ideally, more empirical evidence could be presented to support the ecological validity of these findings.

2. The results of the computational modeling are inconclusive. All three models achieve almost exactly the same fit, with a slight numerical preference for the simplest model. The simpler model might also be preferred on parsimony grounds, as the paper acknowledges. The reasons for nonetheless preferring the more complex PLA model do not strike me as particularly strong. I think it’s fine to report the parameter estimates of that model, as they are interesting even if they are not quantitatively better performing. However, outside of the results section, it feels like too much emphasis is placed on the elements of the PLA model, given that they do not seem to contribute greatly to model fit. For example, the title and abstract led me to expect a much stronger influence

of different learning rates or attribution patterns than actually obtained. I think the conclusions there and in the discussion should be tempered to better reflect the strength of evidence.

3. In Figure 4, it appears that there are very tight 95% confidence intervals around the learning rate and attribution parameters. Given that these parameters seem to differ so much, I find it hard to understand how including them does not substantially improve the model fit of the PLA model above the P-only model. It seems to me that one of two things must be true: either i) those confidence intervals are somehow smaller than they should be, or ii) the inclusion of learning and attribution rates is leading to some form of worse fit that is not being visualized. I would appreciate if the authors could clarify this situation in their revision.

4. Examining the data on OSF (thank you for sharing!) I see that the authors recorded not only the behavioral indexes reported in the paper, but also the fine-grained moves and reaction times of the players throughout the game. I wonder if these data hold clues that might help to resolve some of the ambiguity I discussed in a prior comment, and also strength the paper more generally? For example, if players take circuitous routes to avoid bots, that might provide an additional implicit index of both learning about individuals, and group preferences (particularly for players who are disinclined to zap anyone themselves). I realize that examining these data was not part of the preregistration, and thus would have to be considered as exploratory, but I still think there could be value in doing so.

Mark Thornton

Reviewer #2 (Remarks to the Author):

In this manuscript, Nafcha and Hertz combined computational modeling with an online star-harvest game to test how equal experiences with members of ingroup and outgroup affect impressions about those groups, and how social learning processes support the formation, maintenance, and facilitation of intergroup biases. Reinforcement learning models were applied to examine participants' beliefs about the likelihood of in/outgroup players to zap, and the prior expectation, the learning rates, and the attribution parameters were extracted from the learning model separately for ingroup and outgroup. They found that people have very low learning rates for outgroups compared to ingroups, making it very difficult to overcome prior beliefs about outgroups' zapping behavior. Moreover, they found participants also attributed the negative behavior of one outgroup player to all other outgroup players, making beliefs about outgroup players more homogeneous.

The question of how equal contact experiences with ingroup and outgroup members are integrated and updated is an important and interesting one. Applying reinforcement learning to this topic could provide fresh insight. However, I found it hard to assess the study due to missing information on the model constructions/evaluations. Overall, I think this work has potential but there are many complex aspects to think through.

Major:

1. Lack of detailed information on the decision rule. The authors used the reinforcement learning model to capture participants' beliefs about the likelihood of in/outgroup players executing zap

behavior. All the target parameters, i.e., the priors, learning rates as well as the attribution parameters were based on the learning models, and were estimated separately for the ingroup and the outgroup. However, as far as I understood, they did not measure the actual response reflecting participants' beliefs about others. Instead, they used participants' own zapping behavior as the dependent variable here, based on the assumption that participants were more likely to zap players that displayed zapping behavior. While this assumption is reasonable, I found that there is very little information about the results of the decision rule, despite these being key for all the modeling. What are the statistical results for each weight parameter shown in the decision rule? Did the author use the same or different weight for ingroup/outgroup in the decision rule equation? Currently, they attributed all the intergroup differences to the learning models, but it might be possible that participants learn the behavior of ingroup/outgroup similarly, but they weigh them differently. For example, it could be possible that participants perceive the ingroup/outgroup as equally likely to zap, but they choose to zap the outgroup instead of the ingroup, resulting in different zapping behavior between groups. I think more information as well as a more detailed examination of the decision rule is needed in the current manuscript.

2. Lack of evaluation for the performance of the winning model. The authors only provided the WAIC values for the comparisons between models and they found the WAIC for three candidate models were very similar. Although justification is provided to some extent (Page 16) to select the most complex one, I think it would be necessary to show other justifications for the winning model and to show how well the model could predict the actual behavior of the participant. For example, the author could perform some cross-validation prediction analyses to validate the winning model.

3. On Pages 16-17, the authors only reported the mean value of each modeling parameter, without reporting any statistical results.

4. More analysis on the zap rate measures may be needed. In the section on the model-free analysis (Page 13), the authors examined the aggregated zap rates measures, I think it might be helpful to examine the change of zap rates over time in order to better reflect the "learning mechanisms" (e.g., use the trial index as the predictors, or half-split the trials and compare them against each other), especially when the model-based analysis in the latter section focused on the "learning models".

5. Detailed information on the star-harvest game should be provided in the methods section. For example, how many trials were in the experiment? What is the proportion of the zappers choosing to zap the players? These were important information that need to be mentioned in the methods section.

6. If I understand correctly, the authors used the same learning rate to capture the Group-Attribution effect as well as the learning effect (i.e., Page 9, equations 2 and 3 share the same learning rate [LR_{In/Out}]). Did the author consider using different learning rates here to distinguish the learning effect and group-attribution effect?

Minor:

1. The supplementary information was not well organized. The format of the tables was not matched.

2. In Figure 2, the meaning of the shaded area was not define.

We thank the reviewers for their time and effort and useful comments. Here are our detailed responses to their concerns and suggestions showing how we incorporated them into the revised paper. We especially appreciate the comments about modeling approaches, which we believe greatly helped promote our model fitting and model insights.

For easier reading, we presented the reviewers' comments in black, our responses are presented in blue, and changes to the text are presented in red.

Point-by-point Responses

Reviewer #1 (Remarks to the Author):

Nafcha and Hertz present a behavioral investigation of intergroup bias. In a game paradigm, participants pursued a goal while having potentially hostile interactions (“zapping”) with other (bot) players. Experimental conditions manipulated the group identities of these players (minimal groups or neutral) and their hostile behavior (always zap or never zap). Direct analysis of player behavior showed that participants learned from the bots' behavior, and displayed ingroup bias in terms of how often the players zapped, and how they allocated rewards after the game. Computational modeling investigated three different learning mechanisms to explain this bias: priors, learning rates, and differentiated attributions to other group members. The results showed clear evidence for an effect of priors, but, the results were less conclusive regarding the inclusion of learning rate or attribution effects.

This paper has a number of strengths. It investigates a clear, computationally-specific set of hypotheses in a well-controlled, well-powered, preregistered experiment. The paper is generally clearly written, and investigates a question of social importance. Below are a number of issues which I think it could benefit from addressing further:

1. My main concern with respect to this work is its external/ecological validity. The paper is somewhat unclear about what the players believed regarding the identities of the other players – did they know they were bots, or did they believe they were real people? Even if they were told the bots were real, or if it was left ambiguous, there are a number of clues that might have led participants to believe they were bots. For example, each individual bot played with a completely consistent strategy of zapping or not, which is quite unlike how humans behave in most contexts. If the participants realized/knew the bots were not real people, this calls into question some aspects of the interpretation of their behavior. Specifically, it would make it difficult to know whether the results reflect people's genuine intergroup psychological processes, or the application of a social script specific to competitive games with nonhuman players. Videogaming against computer bot players is now ubiquitous in society, and players are well known to engage in a wide variety of non-naturalistic behaviors to exploit or otherwise manipulate these bots. Moreover, if the participants realized they were playing against bots, they may have attributed the bots behavior to the intentions of the experimenter, raising the risk of compliant demand effects.

This limitation needs to be discussed at greater length in the paper, and ideally, more empirical evidence could be presented to support the ecological validity of these findings.

We agree that ecological validity of our task and result is an important issue. The main reason we use the term ‘bot-players’ throughout the manuscript, is exactly to make sure that readers are cautious when drawing conclusions from our study, and to avoid any misleading conclusions. However, we believe that this concern does not make it impossible to draw any meaningful conclusions, and that participants behaved as if they believe to some extent that they were playing with human participants. We note that our research question is not whether intergroup bias exists, as it is a well-documented phenomenon, but uncovering the underlying learning process, a question where control over participants’ observations and experiences is crucial.

We would like to start by explaining our assumption about the level of participants’ belief about the identity of other players, and how our instructions to participants and the design of the bot-players algorithms support our approach. We will then discuss the reasons for the use of bot-players in our task, and the possible implications of this decision, now explicitly discussed in the main text.

As interaction with anonymous humans and anonymous bots in online multiplayer games and environments become ubiquitous, online participants seem to hold some level of belief that other players may be human (Summerville and Chartier 2013; Thomas and Clifford 2017; Aguinis, Villamor, and Ramani 2021). This was indeed what we found in previous works, where participants playing multiplayer games online were not good at detecting whether they were playing with other human players or algorithmic players (Hertz et al. 2017), and did not differentiate their behavior when playing with bots and humans (Hertz et al. 2017) to the extent that participants were willing to incur monetary loss to gain influence on players whose identity is unknown (Hertz et al. 2020). In fact, results in such multiplayer online settings were highly replicable, even without elaborate cover-story (Zaatri, Aderka, and Hertz 2022). Our aim here was that our participants will maintain some level of belief that other players might be human players.

This affected our instructions to participants - participants were not informed about the identity of the other players in the task, i.e., whether they are other human players or bots. It is true that interacting with bots in online players is ubiquitous, but so is interactions with other anonymous human players in online video games. As the task was carried out online, where interactions with anonymous other humans is plausible and common, we assumed that our participants may be under the impression that other players *may* be humans. This is now explicitly stated in the methods section and task description (page 6 and 27). Second, the design of the algorithms governing the bot players was inspired by the biases observed in participants when designing the bot-players’ algorithms (Hertz 2021)(See figure below, now supplementary figure S1, and revised text in page 6 and 27), and was carried to capture levels of players’ competitiveness Both bot-player types were programmed to first check

whether they are the closest player to a star, and if so to move towards the star, thus concluding their turn. Otherwise, bot-players were programmed to check whether they are in direct competition with another player for a star. This happens when another player is on their way to a star, i.e., closer to the star closest to them. Avoiders were programmed to move away and seek other stars. Zappers were programmed to zap the competing player if they share the same row or column. Zapping is therefore done in the context of competition over a star, and not arbitrarily, i.e., does not occur every time the bot-player *can* zap another player. The two algorithms capture the behavior of star-seeking players with different levels of competitiveness. While this may not completely overcome that problem of participants questioning the identity of other players, we believe it mitigates this problem to some extent, and allows participants to maintain some level of belief that other players may be human as well.

Figure S1 (now in the supplementary materials) – Algorithms governing the zapping behavior of the zap-avoiders (top) and zappers (bottom) bot-players. This figure is adapted from Hertz 2021.

We now turn to the costs and benefits associated with our decision to use bot-players.

Our aim in this study was to investigate the learning dynamics underlying intergroup bias. We were especially interested in the way group identity of different agents shape the way their behavior is perceived and accumulated over time by a learner and affect his behavior. To this end, we needed to make sure that participants interact with players that behave in a similar manner but belong to different reference groups. This could not be achieved using

live interaction, where players' behavior cannot be controlled. Indeed, the exact intergroup bias we report here is predicted to make players from different groups behave very differently, i.e., zapping behavior will be conditioned on group affiliation, directly confounding our research question.

A clear cost for our reliance on bot-players is that it is not clear to what extent our results generalize to real-world interactions. Our behavioral results are that participants differentiate between ingroup and outgroup members in their zapping behavior, and now also in their path-crossing behavior, above and beyond the observed behavior of these players (whether they zap or not). This effect replicates well documented intergroup bias in a variety of experiments and real life situations (Levy et al. 2022; Boyer, Firat, and van Leeuwen 2015; De Dreu, Gross, and Romano 2023; Tajfel et al. 1971). However, it is not clear whether our proposed learning account of the persistence of this bias takes place in real life scenarios. Some evidence suggests that cognitive processes that relate to perception, empathy and impression formation are indeed subject to similar social identity effects (Xu et al. 2009; Shin and Niv 2021; Allidina and Cunningham 2021; Kardosh et al. 2022). These indicate that our mechanism is plausible, and that biases in cognitive processing of other's actions based on their group identity may indeed contribute to some extent to the maintenance of intergroup bias despite similarity in group members behavior.

We revised the discussion to reflect this important limitation (Page 24).

Changes to text (in addition to the inclusion of Figure S1 in the supplementary materials):

Page 6 (introduction):

Participants were not informed about the identity of the other players. Those players were in fact bot-players and were programmed to manifest one of two behavioral patterns. All bot-players were programmed to display star-seeking behavior, and differed in the way they react when another player was standing between them and a star. One type of bot-player zapped this player, clearing the path for them to collect the star, a type called 'zappers'. The other type of bot-player did not zap, and instead moved away towards other stars, a type called 'avoiders' (see methods for full details).

Page 27 (methods):

The behavior of the bot-players was governed by algorithms implementing zapper and avoider behavior, as used in a previous study⁴² (Supplementary Figure S1). Both bot-player types were programmed to first check whether they are the closest player to a star, and if so to move towards the star, thus concluding their turn. Otherwise, bot-players were programmed to check whether they are in direct competition with another player for a star. This happens when another player is on their way to a star, i.e., closer to the star closest to them. Avoiders were programmed to move away and seek other stars. Zappers were programmed to zap the competing player if they share the same row or column. Zapping is therefore done in the context of competition over a star, and not arbitrarily, i.e., does not

occur every time the bot-player *can* zap another player. The two algorithms capture the behavior of star-seeking players with different levels of competitiveness.

Page 24 (discussion):

Several limitations should be considered when interpreting our results. First, we used bot-players to manipulate the behavior the participants were exposed to in the different experimental conditions, which may cast some questions regarding the extent of the ecological validity of this work. Our aim in this work was not to study the formation of group dynamics, such as intergroup bias, but to study the learning mechanisms underlying this well-established phenomenon. This required controlling the behavior of ingroup and outgroup players, that do not display intergroup bias in their behavior and zap/avoid zapping all other players in a similar way, which we achieved by using bot-players. We assumed that in a live interaction setting, participants would have displayed intergroup bias in their zapping behavior, making ingroups and outgroups behave very differently and thus confounding our ability to study the participants learning process. Therefore, our findings regarding learning biases may unfold in a more complex manner in real interactive situations, where in addition to learning biases players actively display discriminatory behavior. We suggest that even in situations where such discrimination may be controlled for, such as in working environments, biased learning may still impact behavior.

It is also important to note that participants were not explicitly told that they were playing with algorithmic players, and it is very likely that they sustained some level of belief that the other players might be other human participants. As interaction with anonymous humans and anonymous bots in online multiplayer games and environments become ubiquitous, and online participants seem to assume that they might be interacting with other participants⁶⁶⁻⁶⁸. In previous works, we found that participants playing multiplayer games online were not good at differentiating between human and algorithmic players⁶⁹, to the extent that participants were willing to incur monetary loss to gain influence on players whose identity is unknown⁷⁰. In addition, the behavior of the algorithmic agents was designed to be similar to human players, as they prioritized star collection over zapping, and zapped during competition over stars and not arbitrarily⁴². Our behavioral results are that participants differentiated between ingroup and outgroup members above and beyond the observed behavior of these players. This effect replicates well documented intergroup bias in a variety of experiments and real life situations^{9,15,71,72}. We therefore suggest that the use of bot-players was not detrimental to the formation of group-identity and intergroup bias.

2. The results of the computational modeling are inconclusive. All three models achieve almost exactly the same fit, with a slight numerical preference for the simplest model. The simpler model might also be preferred on parsimony grounds, as the paper acknowledges. The reasons for nonetheless preferring the more complex PLA model do not strike me as particularly strong. I think it's fine to report the parameter estimates of that model, as they are

interesting even if they are not quantitatively better performing. However, outside of the results section, it feels like too much emphasis is placed on the elements of the PLA model, given that they do not seem to contribute greatly to model fit. For example, the title and abstract led me to expect a much stronger influence of different learning rates or attribution patterns than actually obtained. I think the conclusions there and in the discussion should be tempered to better reflect the strength of evidence.

We thank the reviewer for this comment, which was also raised by the other reviewer, along with some very useful modelling suggestions. Following the reviewers' comments, we revised our modelling approach and used a different formulation of the prior and attribution effects (see more details below). This approach turned out to lead to far more stable fitting than our original formulation in terms of MCMC chains convergence. This was also apparent when we compared the model predictions of zapping behavior. We therefore revised our methods section and uploaded the revised models. In addition, this formulation showed a clear advantage for the PLA model over the other models in terms of WAIC, as can be seen in Table R2. This also resulted in changes to the parameters estimations. We now discuss these thoroughly in the results and discussion sections.

The main changes in model formulation, suggested by reviewer 2, are using group prior parameters outside the learning process, such that learning process tracks only the observed behavior of the players, starting with zap-probability estimation of 0, and group parameters reflect an overall bias towards zapping in/out group even when not evidence for player zap is available.

In this formulation learning process remains the same, but belief at time 0 is changed to 0:

$$\begin{aligned}
 \text{PlayerZap}(0) &= 0 \\
 \text{PlayerZap}(t + 1) &= \text{PlayerZap}(t) + \begin{cases} LR_{Zap}^{In/Out} \cdot (1 - \text{PlayerZap}(t)) \\ LR_{Avoid}^{In/Out} \cdot (0 - \text{PlayerZap}(t)) \end{cases}
 \end{aligned}$$

And the group priors are instead included in the decision rule directly:

$$p(\text{Zap}) = \text{Logistic}(w_0 + \text{Prior}_{in/out} + w_1 \cdot \text{StarDist} + w_s \cdot \text{TargetDist} + w_3 \cdot \text{TargetZap})$$

This means that prior parameters are not bound to be between 0 and 1. It also keeps the interpretation of these variables relatively straightforward, as it indicates the probability of zapping in/outgroup players even when no zaps were observed, i.e., independently of the learning process.

A second suggested change was made to the formulation of the attribution effect. Instead of attribution parameter changing the value of the observed zap/avoidance, it is now represented as the learning rate in which observation of one player is used to update the belief on the other group member zapping behavior:

$$OthersZap(t + 1) = OthersZap(t) + \begin{cases} At_{Zap}^{In/Out} \cdot (1 - OthersZap(t)) \\ At_{Avoid}^{In/Out} \cdot (0 - OthersZap(t)) \end{cases}$$

This makes it easier to compare the attribution to the direct learning mechanisms, and to directly compare the attribution and learning parameters, as they now reflect a similar process.

These changes reflect changes to the *formulation* of the models, and do not change the theoretical building blocks of the models. Importantly, we still have three different mechanisms, prior, learning and attribution, and can compare how these contribute to behavior.

We carried out simulations of the new formulations of the models, showing that we can still capture the three unique contributions. We present simulations carried out using the mixed-behavior structure (zappers and avoiders), where the differentiating contribution of the three models is expected. These effects are presented in the revised Figure 2:

Revised Figure 2 - Model simulations of three learning mechanisms

We simulated the learning model to test how avoiders/zappers players in heterogeneous groups. (A) We disabled learning and attribution effects and set the prior parameters to be either low (-0.9) or high (-0.1). No learning was observed (left panel), and the prior effect dictates zapping behavior estimation regardless of actual zapping behavior (right). Note that situational variables such as distance from target, and the tendency of zappers to be closer to other players, affect simulated zapping behavior. (B) We disabled the attribution effect, fixed the prior effect at 0, and varied the learning rates (LR) of zaps to be either low (0.2) or high (0.9). Learning curves illustrating faster learning of zappers behavior (left),

and difference in zapping rates of zappers and avoiders (right). (C) We fixed learning rates at 0.8, and priors at 0, and varied the zap attribution rate parameters to be either low (0.2) or high (0.5) . High attribution rates increased the estimation of the zapping probability of zap avoiders (left), and led to more similar zapping behavior towards zappers and avoiders (right). Lines in learning curves indicate mean zapping estimation variables, shadows indicate 95% confidence intervals. Boxplots boxes include the median in bold line, interquartile range is represented by the box, minimum and maximum range by the whiskers, and outliers by dots.

We carried the fitting procedure for the three models, and found that the PLA model had the lowest WAIC score (i.e., best fitting to the data):

Model	WAIC	SE	dWAIC	dSE	pWAIC	weight
PLA	4206.67	86.98	0	NA	193.92	0.92
PL	4211.63	87.13	4.95	5.66	178.47	0.08
P	4251.06	87.15	44.39	12.44	155.24	<<0.01

Table 1 – WAIC fitting scores and model comparisons. WAIC scores of each model, sorted from small (better) to large (worse), SE is the standard error of each WAIC, dWAIC is the difference between each model’s score and the best model’s score, dSE is the standard error of this difference, pWAIC is the measure of effective number of parameters, capturing model’s flexibility, and weight is the Aikake weight given to each model in the prediction of participants’ behavior.

This seems to validate our assumption that the PLA model is indeed better supported by participants zapping behavior. The increased stability of fitting procedure helped acquire more reliable and accurate estimations.

Finally, we observed some changes to the values of estimated parameters, now presented in revised figure 5 (see in next comment). We found that prior zapping parameter was lower to in-group compared with out-group players. Learning rates for zaps were higher for in-group compared with outgroups, and in both cases higher than learning rates for zap-avoidance. Attribution rates for zaps and avoidances were similar for in-groups, but attribution of outgroup avoidance was very low. This meant that outgroup zappers did not benefit from the avoidance of their group members, while ingroup zappers did benefit from it, in terms of reduction in zap belief. This resulted in different learning patterns for ingroups and outgroups, and in the different zapping patterns we observed.

Importantly, fitting procedure also indicated a strong effect of estimated target zapping behavior on the likelihood to zap, which indicates that participants were affected by the players’ zapping behavior. We also found that the distance from the target negatively affected likelihood to zap, i.e., participants were less likely to zap players that were far from them, and that being close to stars reduced the likelihood to zap.

Changes in the main text – in addition to the changes to Figure 2 and addition of Table 1 presented above:

Pages 9-11:

Computational Learning Model

First we needed to evaluate how the three learning mechanisms—prior effects, individual learning, and group-level attribution—are captured by a computational model developed in previous work and how they can produce different learning patterns in our experimental design⁴². Our model aimed at identifying the contribution of the three mechanisms to the way participants decide whether or not to zap other players. Our model includes two parts, a decision rule by which participants make the decision to zap other players on a trial-by-trial basis, and a learning mechanism which accumulates the behavior of the other players to establish beliefs about their likelihood to zap. More details regarding the model are included in the methods section.

Decision rule: On a trial-by-trial basis, we modeled the decision to zap or avoid zapping a target player as dependent on a weighted sum of multiple variables (eq. [1]). These variables included the situational factors of distance from the target (weighted by free parameter $w_{DistTarget}$) and distance from the closest star (parameter $w_{DistStar}$). It was also dependent on zap priors: a zapping bias (parameter $Bias$), indicating the overall inclination of participants to zap, and prior related to the group identity of the target, either ingroup or outgroup (parameters $Prior_{in}/Prior_{out}$), indicating the likelihood to zap players based only on their group identity, regardless of their behavior. Finally, the decision was dependent on the learned belief about the target's likelihood to zap (parameter $w_{TargetZap}$). This belief is not immediately available from the data and has to be inferred using the learning part of the model.

$$[1] p(Zap) = \text{Logit}(Bias + Prior_{in}/Prior_{out} + w_{StarDist} \cdot StarDist + w_{TargetDist} \cdot TargetDist + w_{TargetZap} \cdot TargetZap)$$

Learning Mechanism:

To estimate the likelihood of each of the four other players to zap, we used a reinforcement learning mechanism that calculates the prediction error, i.e., the difference between the observed behavior (zap = 1, avoid = 0) on each trial and current estimation of player's likelihood to zap, and updates the belief using a learning rate⁵⁴ (eq. [2]). We assumed different learning rates for avoidance and zapping behavior for in/outgroup members (four free parameters). The initial value for beliefs about players' likelihood to zap was set to 0, as prior beliefs were captured by the $Prior_{in}/Prior_{out}$ parameters in the decision rule.

$$[2] \quad \text{PlayerZap}(t + 1) = \text{PlayerZap}(t) + \begin{cases} LR_{Zap}^{In/Out} \cdot (1 - \text{PlayerZap}(t)) \\ LR_{Avoid}^{In/Out} \cdot (0 - \text{PlayerZap}(t)) \end{cases}$$

Group-Attribution was modeled in terms of updating the beliefs not only about the player that just acted, but also about his other group members. We assumed different group-attribution values for avoidance and zapping behavior for in/outgroup members (four free parameters). These free parameters governed the rate of update, similar to learning rates.

$$[3] \quad \text{OthersZap}(t + 1) = \text{OthersZap}(t) + \begin{cases} At_{Zap}^{In/Out} \cdot (1 - \text{OthersZap}(t)) \\ At_{Avoid}^{In/Out} \cdot (0 - \text{OthersZap}(t)) \end{cases}$$

Model Simulations

To examine our model and experimental design ability to capture the effects of different learning mechanisms, we simulate the model using data collected in a pilot study (see supplementary materials). For the model simulations, we discarded the participants' behavior and used only the bot-players' behavior (location and zaps) and the star locations to generate expected zapping behavior under different experimental conditions. In each simulation, we either disabled two learning mechanisms or kept them fixed and varied the model parameters associated with the mechanism of interest. We examined the learning patterns in the heterogeneous conditions, where different contributions from the learning mechanisms was expected to be most pronounced. We retrieved the trial-by-trial beliefs about players' zap probabilities estimated by the model, and the zapping rate towards each player (Figure 2).

We began examining how prior effects shape beliefs and zap behavior by setting learning and attribution rates to zero. We used two values for priors—low (-0.9) and high (-0.1). The model's estimation of beliefs about players' zap probabilities did not change, regardless of the players' zap behavior or the behavior of their group members (Figure 2A). The zapping pattern was dependent on the priors' value, with a higher likelihood to zap when prior value was high compared to low.

We then examined how learning rate values shape learning by fixing the prior parameters and attribution-rates to 0. We also kept the avoidance learning rate fixed at 0.25, and zap learning rate to be either low (0.2) or high (0.9). The estimation of beliefs about players' zapping probability was dependent on the players' behavior, increasing only for zappers and not for avoiders. The model predicted faster learning and higher zapping beliefs

when learning rates were high. These beliefs were translated to higher zapping rates for zappers compared with avoiders, and higher zapping rates when learning rates were high.

Finally, to evaluate the group-level attribution effects, we kept priors fixed at 0, zapping learning rates fixed at 0.8 and avoidance learning rates fixed at 0.25. We also fixed the avoidance attribution rates at 0, and set the zapping attribution rates to be either low (0.2) or high (0.5). The model predicted that beliefs about players' zapping behavior would be higher for zappers compared with avoiders, but that estimation of avoiders' likelihood to zap would increase over time even though they never zapped, due to the group-level attribution. High attribution rates led to increased similarity in beliefs about players' zapping behavior between the zappers and avoiders. This translated to increased similarity in zapping rates of zappers and avoiders when attribution rates were high.

Page 18 (Model Fitting Results):

We fitted three different models: The first included only the prior mechanism (P model, eq [1]), the second included prior and learning mechanisms (PL model, eq [1] +eq [2]), and the third included prior, learning and group-level attribution mechanisms (PLA model, eq [1] +eq [2] +eq [3]). Our model fitting and model comparison procedure yielded the lowest WAIC fitting score to the PLA model, indicating that it best described the participants' decisions while accounting for its increased complexity and number of parameters (Table 1). Support for the PLA model was already indicated in the aggregated behavioral results, which demonstrated prior, learning and group-level attribution effects.

3. In Figure 4, it appears that there are very tight 95% confidence intervals around the learning rate and attribution parameters. Given that these parameters seem to differ so much, I find it hard to understand how including them does not substantially improve the model fit of the PLA model above the P-only model. It seems to me that one of two things must be true: either i) those confidence intervals are somehow smaller than they should be, or ii) the inclusion of learning and attribution rates is leading to some form of worse fit that is not being visualized. I would appreciate if the authors could clarify this situation in their revision.

We thank the reviewer for pointing this out. We plotted the mean and confidence intervals of the mean from the individual-level parameters. In our hierarchical model fitting, the individual-level parameters are calculated by adding an individual value to the group-level parameters, and the individual-level values distributed around the group-level parameter. We agree that this presentation is misleading, as it does not convey the uncertainty regarding the

group-level parameters. We corrected this problem, and now plot the posterior distribution of the group-level parameters, i.e., the distribution of parameters' values across the MCMC chains during the sampling portion of the model fitting procedure. We also use boxplot instead of mean and confidence interval, to better convey the shape of the distribution. Below is our revised Figure 5, with the new parameters values following the revised modelling approach. Finally, we included all group-level parameters mean and 89% HDI (high density interval, where 89% of the distribution is located) in the text and in Table 2.

Changes to the text:

A. Estimated Model Parameters

B. Estimated Learning Pattern

C. Estimated Zap Pattern

Revised Figure 5 – PLA Social learning model results

We estimated group level parameters of the PLA social learning model. (A) Posterior distribution of the model group-level parameter. 1. Priors include the general bias parameter, and in-group and out-group priors. Out-group prior to zap were higher than in-group prior. 2. Learning rates were higher for zaps compared with avoidances, and were highest for out-group zaps. 3. Attribution rates for out-group avoidance were close to zero, whereas in all other cases they were higher. 4. Situational parameters indicate that likelihood to zap increased when stars were far away, decreased when the target player was far away, and increased when the estimation of the target player's likelihood to zap was high. (B) Model estimations of the internal learning process about player's likelihood to zap in the different experimental conditions. Higher learning rates for outgroup zaps, and lower attribution rates for outgroup avoidances contributed to higher estimation of zap probability both for outgroup zappers and avoiders. (C) Model predicted zapping behavior shows a group and behavior effect, like the observed behavioral pattern. Lines in learning curves indicate mean zapping estimation variables, shadows indicate 95% confidence intervals. Boxplots boxes include the median in bold line, interquartile range is represented by the box, minimum and maximum range by the whiskers, and outliers by dots.

Pages 19-20 (Results):

Our PLA model fitting procedure resulted in estimation of the posterior of group-level parameters, allowing us to compare the learning mechanisms in learning about the zapping behavior of ingroup and outgroup players (Figure 5A, Table 2). For the decision rule (eq. [1]), we found a support for a positive effect of distance from stars ($w_{DistStars}$ mean: 0.26, 89% high density interval (HDI): [-0.04, 0.58]), as participants were more likely to zap while stars far from them. We found a strong support for a negative distance from target-player stars ($w_{DistTarget}$ mean: -1.84, 89% HDI: [-2.23, -1.46]), as participants were more likely to zap target-players that were close to them. Importantly, we found a strong support for the effect of the target player's likelihood to zap on participants' zapping decisions ($w_{TargetZap}$ mean: 0.89, 89% HDI: [0.53, 1.22]). Finally, an overall negative zap-bias was observed ($bias$ mean: -1.01, 89% HDI: [-2.86, 0.90]), indicating that participants tended to avoid zapping overall.

Variable	mean	median	sd	rhat	89% HDI
Prior_In	-0.87	-0.87	1.17	1.0005	[-2.76, 0.98]
Prior_Out	-0.12	-0.13	1.17	1.0005	[-1.99, 1.76]
LR_Zap_In	0.71	0.70	0.84	1.0016	[0.27, 1.00]
LR_Zap_Out	0.84	0.82	0.78	1.0009	[0.55, 1.00]
LR_Avoid_In	0.08	0.07	0.85	1.0020	[0.00, 0.42]
LR_Avoid_Out	0.13	0.13	0.69	1.0000	[0.02, 0.27]
Bias	-1.01	-1.01	1.17	1.0005	[-2.86, 0.90]
w_dist_star	0.03	0.03	0.02	1.0001	[-0.04, 0.58]
w_dist_Target	-0.18	-0.18	0.02	0.9998	[-2.23, -1.46]
w_zap_Target	0.89	0.88	0.23	1.0007	[0.53, 1.22]
At_Zap_In	0.19	0.19	0.83	1.0004	[0.00, 0.60]
At_Zap_Out	0.22	0.22	0.75	1.0009	[0.01, 0.48]
At_Avoid_In	0.14	0.13	0.80	1.0008	[0.00, 0.44]
At_Avoid_Out	0.02	0.03	0.73	1.0004	[0.00, 0.06]

Table 2: Parameter Estimations of the computational learning model. Mean, median and standard error (sd) of the posterior distribution of group level parameters is presented, along with the 89% HDI. Rhat represents the convergence of the mcmc chain, with values close to 1 representing convergence.

We then examined group-identity dependent differences in parameter values. Group identity prior parameters were different from each other, as ingroup prior was lower than outgroup prior ($Prior_{out} - Prior_{in}$ mean: 0.75, 89% HDI: [0.55, 0.96]). Learning rates were overall higher for zaps than for avoidances within each group ($LR_{zap}^{in} - LR_{avoid}^{in}$ mean: 0.49, 89% HDI: [0.03, 0.99], $LR_{zap}^{out} - LR_{avoid}^{out}$ mean: 0.63, 89% HDI: [0.35, 0.92]), in line with previous works. Learning rates for zaps tended to be higher for outgroup compared with ingroups

($LR_{zap}^{out} - LR_{zap}^{in}$ mean: 0.13, 89% HDI: [-0.27, 0.60]), and learning rates for avoidances were similar between groups ($LR_{avoid}^{out} - LR_{avoid}^{in}$ mean: 0, 89% HDI: [-0.34, 0.31]). Finally, attribution rates for zaps and avoids were similar for ingroups ($At_{zap}^{in} - At_{avoid}^{in}$ mean: 0.06, 89% HDI: [-0.41, 0.67]), but were different for outgroups ($At_{zap}^{out} - At_{avoid}^{out}$ mean: 0.22, 89% HDI: [-0.04, 0.49]), and attribution for avoidances was higher for ingroups compared with outgroups ($At_{avoid}^{in} - At_{avoid}^{out}$ mean: 0.16, 89% HDI: [-0.06, 0.44]), as attribution rates for outgroup avoidances were very low. These results indicate three group-identity asymmetries in learning parameters. First, priors for zaps were higher for outgroup compared with ingroup target players. Second, learning rates for zaps were higher when observing outgroup bot-players compared with ingroup bot-players. Third, group-level attribution of avoidances was lower when observing outgroup bot-players than ingroup bot-players.

4. Examining the data on OSF (thank you for sharing!) I see that the authors recorded not only the behavioral indexes reported in the paper, but also the fine-grained moves and reaction times of the players throughout the game. I wonder if these data hold clues that might help to resolve some of the ambiguity I discussed in a prior comment, and also strengthen the paper more generally? For example, if players take circuitous routes to avoid bots, that might provide an additional implicit index of both learning about individuals, and group preferences (particularly for players who are disinclined to zap anyone themselves). I realize that examining these data was not part of the preregistration, and thus would have to be considered as exploratory, but I still think there could be value in doing so.

We thank the reviewer for these suggestions, we found them very useful both for providing supporting evidence for intergroup bias and learning, and for demonstrating the richness of the data obtained in our experimental approach.

We examined the likelihood of participants to cross-path with another player, i.e., to choose to move to a location where they share the same row or column with another player. Cross-pathing means that the participant becomes vulnerable as the other player can zap them in the next trial. We used this measure as a dependent variable in a mixed-effects linear regression, which included group identity (ingroup/outgroup/neutral), bot-player's behavior (zapper/avoider), and group homogeneity (homogeneous/heterogeneous), and the interactions between these factors. Our analysis revealed a significant effect of group identity ($F_{(2, 1012)} = 6.65, p = 0.0013, \eta_{partial} = 0.013$) and a significant effect of bot-player behavior ($F_{(1, 1983.5)} = 4.21, p = 0.04, \eta_{partial} = 0.002$). Participants were more likely to cross-path with in-groups, and avoid outgroups, indicating an intergroup bias in their estimation of the risk posed by

ingroup and outgroup players. In addition, participants were sensitive to the behavior of the players, and were more likely to avoid zappers and to cross-path with zap-avoiders.

Panel A of revised Figure 3: Path-cross frequency, indicating the frequency of participants deciding to move to a position where another player can zap them, i.e., where they share row or column with another player. Participants were more likely to cross-path with ingroup players, and with zap-avoiders.

We also examined the response time of participants when deciding to zap other players. We used the same mixed-effect linear regression, with response time as dependent variable. We found a significant effect of bot-player behavior ($F_{(1,1281.3)} = 10.017, p = 0.0016, \eta_{\text{partial}}=0.0077$), as participants were slower to zap players that were zap-avoiders. We also found a significant interaction between group identity and bot-player behavior ($F_{(2,1271.61)} = 7.99, p = 0.0003, \eta_{\text{partial}}=0.012$), as participants were very slow to zap ingroup zap-avoiders, especially in the homogenous condition (see figure below).

However, we decided not to include this analysis in the manuscript, as the number of zaps was dramatically different between conditions and targets, and therefore the number of observed response times was different between conditions and targets, making this variable less reliable.

Zap response time in the different experimental conditions. Participants were more likely to cross-path with ingroup players, and with zap-avoiders.

Changes to text:

Pages 15-16:

We examined several other behavioral measures that can indicate intergroup bias in our experiment (Figure 4 and supplementary materials). In an exploratory analysis, we examined the likelihood of participants to cross-path with another player, i.e., to choose to move to a location where they share the same row or column with another player. Cross-pathing means that the participant becomes vulnerable as the other player can zap them in the next trial. We used this measure as a dependent variable in a mixed-effects linear regression, which included group identity (ingroup/outgroup/neutral), bot-player's behavior (zapper/avoider), and group homogeneity (homogeneous/heterogeneous), and the interactions between these factors as independent variables. Our analysis revealed a significant effect of group identity ($F(2, 1012) = 6.65, p = 0.0013, \eta_{\text{partial}} = 0.013$) and a significant effect of bot-player behavior ($F(1, 1983.5) = 4.21, p = 0.04, \eta_{\text{partial}} = 0.002$) (Table ST5 in the supplementary materials).

Participants were more likely to cross-path with in-groups, and avoid outgroups, indicating an intergroup bias in their estimation of the risk posed by ingroup and outgroup players. In addition, participants were sensitive to the behavior of the players, and were more likely to avoid zappers and to cross-path with zap-avoiders.

Figure 4: Additional behavioral manifestations of intergroup bias. (A) Path-cross frequency, indicating the frequency of participants deciding to move to a position where another player can zap them. Participants were more likely to cross-path with ingroup than outgroup players, and with avoiders compared with zappers. (B) After the experiment, participants were asked to distribute an extra ten stars between bot-players. They allocated more stars to ingroup bot-players than to outgroup players. In heterogenous conditions they allocated more stars to zap-avoiders, except in the outgroup condition, in which both group members were treated similarly. Boxplots include the mean by light circles, median in bold line, interquartile range is represented by the box, minimum and maximum range by the whiskers, and outliers by black dots.

Reviewer #2 (Remarks to the Author):

In this manuscript, Nafcha and Hertz combined computational modeling with an online star-harvest game to test how equal experiences with members of ingroup and outgroup affect impressions about those groups, and how social learning processes support the formation, maintenance, and facilitation of intergroup biases. Reinforcement learning models were applied to examine participants' beliefs about the likelihood of in/outgroup players to zap, and the prior expectation, the learning rates, and the attribution parameters were extracted from the learning model separately for ingroup and outgroup. They found that people have very low learning rates for outgroups compared to ingroups, making it very difficult to overcome prior beliefs about outgroups' zapping behavior. Moreover, they found participants also attributed the negative behavior of one outgroup player to all other outgroup players, making beliefs about outgroup players more homogeneous.

The question of how equal contact experiences with ingroup and outgroup members are integrated and updated is an important and interesting one. Applying reinforcement learning to this topic could provide fresh insight. However, I found it hard to assess the study due to missing information on the model constructions/evaluations. Overall, I think this work has potential but there are many complex aspects to think through.

Major:

1. Lack of detailed information on the decision rule. The authors used the reinforcement learning model to capture participants' beliefs about the likelihood of in/outgroup players executing zap behavior. All the target parameters, i.e., the priors, learning rates as well as the attribution parameters were based on the learning models, and were estimated separately for the ingroup and the outgroup. However, as far as I understood, they did not measure the actual response reflecting participants' beliefs about others. Instead, they used participants' own zapping behavior as the dependent variable here, based on the assumption that participants were more likely to zap players that displayed zapping behavior. While this assumption is reasonable, I found that there is very little information about the results of the decision rule, despite these being key for all the modeling. What are the statistical results for each weight parameter shown in the decision rule? Did the author use the same or different weight for ingroup/outgroup in the decision rule equation? Currently, they attributed all the

intergroup differences to the learning models, but it might be possible that participants learn the behavior of ingroup/outgroup similarly, but they weigh them differently. For example, it could be possible that participants perceive the ingroup/outgroup as equally likely to zap, but they choose to zap the outgroup instead of the ingroup, resulting in different zapping behavior between groups. I think more information as well as a more detailed examination of the decision rule is needed in the current manuscript.

First, we want to thank the reviewer for his suggestion about the different formulation of our model, where the priors to zap are separated from the learning process. We used this formulation and found that it yielded a much more stable model fit and convergence.

In the revised model, the decision to zap is related to the prior or bias parameters (a general bias and the in/out group prior), the distance from stars and the target player, and the learned likelihood of the target player to zap:

$$p(\text{Zap}) = \text{Logistic}(w_0 + \text{Prior}_{\text{in/out}} + w_1 \cdot \text{StarDist} + w_s \cdot \text{TargetDist} + w_3 \cdot \text{TargetZap})$$

This is the decision rule, and the model is optimized to fit these estimated zap probabilities to the actual zapping behavior of the participants.

The learning process is now independent, and initiates with 0 likelihood to zap. Note that the initial value of 0 was chosen in order to make the value of priors easier to interpret, as it indicates the likelihood to zap a player with no evidence of zapping behavior.

$$\text{PlayerZap}(0) = 0$$

$$\text{PlayerZap}(t + 1) = \text{PlayerZap}(t) + \begin{cases} LR_{\text{Zap}}^{\text{In/Out}} \cdot (1 - \text{PlayerZap}(t)) \\ LR_{\text{Avoid}}^{\text{In/Out}} \cdot (0 - \text{PlayerZap}(t)) \end{cases}$$

A second suggested change in later comment was made to the formulation of the attribution effect. Instead of attribution parameter changing the value of the observed zap/avoidance, it is now represented as the learning rate in which observation of one player is used to update the belief on the other group member zapping behavior:

$$\text{OthersZap}(t + 1) = \text{OthersZap}(t) + \begin{cases} At_{\text{Zap}}^{\text{In/Out}} \cdot (1 - \text{OthersZap}(t)) \\ At_{\text{Avoid}}^{\text{In/Out}} \cdot (0 - \text{OthersZap}(t)) \end{cases}$$

We observed some changes to the values of estimated parameters, now presented in revised figure 5 (see in next comment). We found that prior zapping parameter was lower to in-group compared with out-group players. Learning rates for zaps were higher for in-group compared with outgroups, and in both cases higher than learning rates for zap-avoidance. Attribution rates for zaps and avoidances were similar for in-groups, but attribution of outgroup avoidance was very low. This meant that outgroup zappers did not benefit from the avoidance of their group members, while ingroup zappers did benefit from it, in terms of reduction in zap belief. This resulted in different learning patterns for ingroups and outgroups, and in the different zapping patterns we observed.

Importantly, fitting procedure also indicated a strong effect of estimated target zapping behavior on the likelihood to zap, which indicates that participants were affected by the players' zapping behavior. We also found that the distance from the target negatively affected likelihood to zap, i.e., participants were less likely to zap players that were far from them, and that being close to stars reduced the likelihood to zap.

We now include the posterior distribution of all group-level parameters, including the decision weights, in the revised figure 5 and supplementary table T2.

In addition, we present information about the distribution of the differences between parameters to evaluate how different they are from each other. Note that a simple t-test between the posterior distributions may show inflated results due to the large number of iterations in the MCMC chains. We therefore included only the 89%HDI (high density interval) of the paired differences between parameters, and refrained from claims of significance differences. We believe that this best represents the notion that the combined contribution of the differences in parameters is what underlies the observed behavioral effects.

Changes to main text:

Pages 9-10 (model description):

Our model aimed at identifying the contribution of the three mechanisms to the way participants decide whether or not to zap other players. Our model includes two parts, a decision rule by which participants make the decision to zap other players on a trial-by-trial basis, and a learning mechanism which accumulates the behavior of the other players to establish beliefs about their likelihood to zap. More details regarding the model are included in the methods section.

Decision rule: On a trial-by-trial basis, we modeled the decision to zap or avoid zapping a target player as dependent on a weighted sum of multiple variables (eq. [1]). These variables included the situational factors of distance from the target (weighted by free parameter $w_{DistTarget}$) and distance from the closest star (parameter $w_{DistStar}$). It was also dependent on zap priors: a zapping bias (parameter $Bias$), indicating the overall inclination of participants to zap, and prior related to the group identity of the target, either ingroup or outgroup (parameters $Prior_{in}/Prior_{out}$), indicating the likelihood to zap players based only on their group identity, regardless of their behavior. Finally, the decision was dependent on the learned belief about the target's likelihood to zap (parameter $w_{TargetZap}$). This belief is not immediately available from the data and has to be inferred using the learning part of the model.

$$[1] \quad p(\text{Zap}) = \text{Logit}(\text{Bias} + \text{Prior}_{in}/\text{Prior}_{out} + w_{\text{StarDist}} \cdot \text{StarDist} + w_{\text{TargetDist}} \cdot \text{TargetDist} + w_{\text{TargetZap}} \cdot \text{TargetZap})$$

Learning Mechanism:

To estimate the likelihood of each of the four other players to zap, we used a reinforcement learning mechanism that calculates the prediction error, i.e., the difference between the observed behavior (zap = 1, avoid = 0) on each trial and current estimation of player's likelihood to zap, and updates the belief using a learning rate⁵⁴ (eq. [2]). We assumed different learning rates for avoidance and zapping behavior for in/outgroup members (four free parameters). The initial value for beliefs about players' likelihood to zap was set to 0, as prior beliefs were captured by the $\text{Prior}_{in}/\text{Prior}_{out}$ parameters in the decision rule.

$$[2] \quad \text{PlayerZap}(t + 1) = \text{PlayerZap}(t) + \begin{cases} LR_{\text{Zap}}^{In/Out} \cdot (1 - \text{PlayerZap}(t)) \\ LR_{\text{Avoid}}^{In/Out} \cdot (0 - \text{PlayerZap}(t)) \end{cases}$$

Group-Attribution was modeled in terms of updating the beliefs not only about the player that just acted, but also about his other group members. We assumed different group-attribution values for avoidance and zapping behavior for in/outgroup members (four free parameters). These free parameters governed the rate of update, similar to learning rates.

$$[3] \quad \text{OthersZap}(t + 1) = \text{OthersZap}(t) + \begin{cases} At_{\text{Zap}}^{In/Out} \cdot (1 - \text{OthersZap}(t)) \\ At_{\text{Avoid}}^{In/Out} \cdot (0 - \text{OthersZap}(t)) \end{cases}$$

Pages 19-22 (model fitting results):

Our PLA model fitting procedure resulted in estimation of the posterior of group-level parameters, allowing us to compare the learning mechanisms in learning about the zapping behavior of ingroup and outgroup players (Figure 5A, Table 2). For the decision rule (eq. [1]), we found a support for a positive effect of distance from stars ($w_{\text{DistStars}}$ mean: 0.26, 89% high density interval (HDI): [-0.04, 0.58]), as participants were more likely to zap while stars far from them. We found a strong support for a negative distance from target-player stars ($w_{\text{DistTarget}}$ mean: -1.84, 89% HDI: [-2.23, -1.46]), as participants were more likely to zap target-players that were close to them. Importantly, we found a strong support for the effect of the target player's likelihood to zap on participants' zapping decisions ($w_{\text{TargetZap}}$ mean: 0.89, 89% HDI: [0.53, 1.22]). Finally, an overall negative zap-bias was observed (bias mean: -1.01, 89% HDI: [-2.86, 0.90]), indicating that participants tended to avoid zapping overall.

Variable	mean	median	sd	rhat	89% HDI
Prior_In	-0.87	-0.87	1.17	1.0005	[-2.76, 0.98]
Prior_Out	-0.12	-0.13	1.17	1.0005	[-1.99, 1.76]
LR_Zap_In	0.71	0.70	0.84	1.0016	[0.27, 1.00]
LR_Zap_Out	0.84	0.82	0.78	1.0009	[0.55, 1.00]
LR_Avoid_In	0.08	0.07	0.85	1.0020	[0.00, 0.42]
LR_Avoid_Out	0.13	0.13	0.69	1.0000	[0.02, 0.27]
Bias	-1.01	-1.01	1.17	1.0005	[-2.86, 0.90]
w_dist_star	0.03	0.03	0.02	1.0001	[-0.04, 0.58]
w_dist_Target	-0.18	-0.18	0.02	0.9998	[-2.23, -1.46]
w_zap_Target	0.89	0.88	0.23	1.0007	[0.53, 1.22]
At_Zap_In	0.19	0.19	0.83	1.0004	[0.00, 0.60]
At_Zap_Out	0.22	0.22	0.75	1.0009	[0.01, 0.48]
At_Avoid_In	0.14	0.13	0.80	1.0008	[0.00, 0.44]
At_Avoid_Out	0.02	0.03	0.73	1.0004	[0.00, 0.06]

Table 2: Parameter Estimations of the computational learning model. Mean, median and standard error (sd) of the posterior distribution of group level parameters is presented, along with the 89% HDI. Rhat represents the convergence of the mcmc chain, with values close to 1 representing convergence.

We then examined group-identity dependent differences in parameter values. Group identity prior parameters were different from each other, as ingroup prior was lower than outgroup prior ($Prior_{out} - Prior_{in}$ mean: 0.75, 89% HDI: [0.55, 0.96]). Learning rates were overall higher for zaps than for avoidances within each group ($LR_{zap}^{in} - LR_{avoid}^{in}$ mean: 0.49, 89% HDI: [0.03, 0.99], $LR_{zap}^{out} - LR_{avoid}^{out}$ mean: 0.63, 89% HDI: [0.35, 0.92]), in line with previous works. Learning rates for zaps tended to be higher for outgroup compared with ingroups ($LR_{zap}^{out} - LR_{zap}^{in}$ mean: 0.13, 89% HDI: [-0.27, 0.60]), and learning rates for avoidances were similar between groups ($LR_{avoid}^{out} - LR_{avoid}^{in}$ mean: 0, 89% HDI: [-0.34, 0.31]). Finally, attribution rates for zaps and avoids were similar for ingroups ($At_{zap}^{in} - At_{avoid}^{in}$ mean: 0.06, 89% HDI: [-0.41, 0.67]), but were different for outgroups ($At_{zap}^{out} - At_{avoid}^{out}$ mean: 0.22, 89% HDI: [-0.04, 0.49]), and attribution for avoidances was higher for ingroups compared with outgroups ($At_{avoid}^{in} - At_{avoid}^{out}$ mean: 0.16, 89% HDI: [-0.06, 0.44]), as attribution rates for outgroup avoidances were very low. These results indicate three group-identity asymmetries in learning parameters. First, priors for zaps were higher for outgroup compared with ingroup target players. Second, learning rates for zaps were higher when observing outgroup bot-players compared with ingroup bot-players. Third, group-level attribution of avoidances was lower when observing outgroup bot-players than ingroup bot-players.

Figure 5: Social learning model results We estimated group level parameters of the PLA social learning model. (A) Posterior distribution of the model group-level parameter. 1. Priors include the general bias parameter, and ingroup and outgroup priors. Out-group prior to zap were higher than in-group prior. 2. Learning rates were higher for zaps compared with avoidances and were highest for outgroup zaps. 3. Attribution rates for outgroup avoidance were lower than all other attribution rates. 4. Situational parameters indicate that likelihood to zap increased when stars were far, decreased when the target player was far, and increased when the estimation of the target player’s likelihood to zap was high. (B) Model estimations of the internal learning process about bot-players’ likelihood to zap in the different experimental conditions. Higher learning rates for outgroup zaps, and lower attribution rates for outgroup avoidances contributed to higher estimation of zap probability both for outgroup zappers and avoiders. (C) Model predicted zapping behavior shows a group and behavior effect, like the observed behavioral pattern. Lines in learning curves indicate mean zapping estimation variables, shadows indicate 95% confidence intervals. Boxplots include the median in bold line, interquartile range is represented by the box, minimum and maximum range by the whiskers, and outliers by dots.

To examine how these parameter values shaped learning about other players’ zap behavior, we used the mean parameter estimations to simulate the model in all experimental conditions (Figure 5). We found that our model was able to capture the pattern of participants’ zapping frequency. We then inspected the underlying learning curves about the likelihood of the bot-players’ likelihood to zap. We found that outgroup’s zapping behavior was learned faster, and reached higher plateau, in line with the higher learning rates for outgroups’ zaps. We also found that for outgroups, zappers’ likelihood to zap did not decrease in the mixed condition, while avoiders’ likelihood to zap increased, in line with the asymmetric attribution rates for outgroups. Overall, these results show how asymmetric evidence accumulation and

priors make outgroup players appear more competitive, and make intergroup bias persist even when the behavior of players is readily available.

2. Lack of evaluation for the performance of the winning model. The authors only provided the WAIC values for the comparisons between models and they found the WAIC for three candidate models were very similar. Although justification is provided to some extent (Page 16) to select the most complex one, I think it would be necessary to show other justifications for the winning model and to show how well the model could predict the actual behavior of the participant. For example, the author could perform some cross-validation prediction analyses to validate the winning model.

We carried the fitting procedure for the *revised* three models (detailed above) and found that the PLA model had the lowest WAIC score (i.e., best fitting to the data). We also compared the models in their contribution to the prediction of behavior and found that PLA model was given 92% weight in prediction of behavior.

Changes to text:

Pages 18-19, including table 1:

We fitted three different models: The first included only the prior mechanism (P model, eq [1]), the second included prior and learning mechanisms (PL model, eq [1] +eq [2]), and the third included prior, learning and group-level attribution mechanisms (PLA model, eq [1] +eq [2] +eq [3]). Our model fitting and model comparison procedure yielded the lowest WAIC fitting score to the PLA model, indicating that it best described the participants' decisions while accounting for its increased complexity and number of parameters (Table 1). Support for the PLA model was already indicated in the aggregated behavioral results, which demonstrated prior, learning and group-level attribution effects.

Model	WAIC	SE	dWAIC	dSE	pWAIC	weight
PLA	4206.67	86.98	0	NA	193.92	0.92
PL	4211.63	87.13	4.95	5.66	178.47	0.08
P	4251.06	87.15	44.39	12.44	155.24	<<0.01

Table 1 – WAIC fitting scores and model comparisons. WAIC scores of each model, sorted from small (better) to large (worse), SE is the standard error of each WAIC, dWAIC is the difference between each model's score and the best model's score, dSE is the standard error of this difference, pWAIC is the measure of effective number of parameters, capturing model's flexibility, and weight is the Aikake weight given to each model in the prediction of participants' behavior⁵⁵.

3. On Pages 16-17, the authors only reported the mean value of each modeling parameter, without reporting any statistical results.

We now include the posterior distribution of all group-level parameters, including the decision weights, in the revised figure 5 and supplementary table T2.

In addition, we present information about the distribution of the differences between parameters to evaluate how different they are from each other. Note that a simple t-test between the posterior distributions may show inflated results due to the large number of iterations in the MCMC chains. We therefore included only the 89% HDI (high density interval) of the paired differences between parameters, and refrained from claims of significance differences. We believe that this best represents the notion that the combined contribution of the differences in parameters is what underlies the observed behavioral effects.

Changes to text:

Pages 19-22 (model fitting results):

Our PLA model fitting procedure resulted in estimation of the posterior of group-level parameters, allowing us to compare the learning mechanisms in learning about the zapping behavior of ingroup and outgroup players (Figure 5A, Table 2). For the decision rule (eq. [1]), we found a support for a positive effect of distance from stars ($w_{DistStars}$ mean: 0.26, 89% high density interval (HDI): [-0.04, 0.58]), as participants were more likely to zap while stars far from them. We found a strong support for a negative distance from target-player stars ($w_{DistTarget}$ mean: -1.84, 89% HDI: [-2.23, -1.46]), as participants were more likely to zap target-players that were close to them. Importantly, we found a strong support for the effect of the target player's likelihood to zap on participants' zapping decisions ($w_{TargetZap}$ mean: 0.89, 89% HDI: [0.53, 1.22]). Finally, an overall negative zap-bias was observed ($bias$ mean: -1.01, 89% HDI: [-2.86, 0.90]), indicating that participants tended to avoid zapping overall.

Variable	mean	median	sd	rhat	89% HDI
Prior_In	-0.87	-0.87	1.17	1.0005	[-2.76, 0.98]
Prior_Out	-0.12	-0.13	1.17	1.0005	[-1.99, 1.76]
LR_Zap_In	0.71	0.70	0.84	1.0016	[0.27, 1.00]
LR_Zap_Out	0.84	0.82	0.78	1.0009	[0.55, 1.00]
LR_Avoid_In	0.08	0.07	0.85	1.0020	[0.00, 0.42]
LR_Avoid_Out	0.13	0.13	0.69	1.0000	[0.02, 0.27]
Bias	-1.01	-1.01	1.17	1.0005	[-2.86, 0.90]
w_dist_star	0.03	0.03	0.02	1.0001	[-0.04, 0.58]
w_dist_Target	-0.18	-0.18	0.02	0.9998	[-2.23, -1.46]

w_zap_Target	0.89	0.88	0.23	1.0007	[0.53, 1.22]
At_Zap_In	0.19	0.19	0.83	1.0004	[0.00, 0.60]
At_Zap_Out	0.22	0.22	0.75	1.0009	[0.01, 0.48]
At_Avoid_In	0.14	0.13	0.80	1.0008	[0.00, 0.44]
At_Avoid_Out	0.02	0.03	0.73	1.0004	[0.00, 0.06]

Table 2: Parameter Estimations of the computational learning model. Mean, median and standard error (sd) of the posterior distribution of group level parameters is presented, along with the 89% HDI. Rhat represents the convergence of the mcmc chain, with values close to 1 representing convergence.

We then examined group-identity dependent differences in parameter values. Group identity prior parameters were different from each other, as ingroup prior was lower than outgroup prior ($Prior_{out} - Prior_{in}$ mean: 0.75, 89% HDI: [0.55, 0.96]). Learning rates were overall higher for zaps than for avoidances within each group ($LR_{zap}^{in} - LR_{avoid}^{in}$ mean: 0.49, 89% HDI: [0.03, 0.99], $LR_{zap}^{out} - LR_{avoid}^{out}$ mean: 0.63, 89% HDI: [0.35, 0.92]), in line with previous works. Learning rates for zaps tended to be higher for outgroup compared with ingroups ($LR_{zap}^{out} - LR_{zap}^{in}$ mean: 0.13, 89% HDI: [-0.27, 0.60]), and learning rates for avoidances were similar between groups ($LR_{avoid}^{out} - LR_{avoid}^{in}$ mean: 0, 89% HDI: [-0.34, 0.31]). Finally, attribution rates for zaps and avoids were similar for ingroups ($At_{zap}^{in} - At_{avoid}^{in}$ mean: 0.06, 89% HDI: [-0.41, 0.67]), but were different for outgroups ($At_{zap}^{out} - At_{avoid}^{out}$ mean: 0.22, 89% HDI: [-0.04, 0.49]), and attribution for avoidances was higher for ingroups compared with outgroups ($At_{avoid}^{in} - At_{avoid}^{out}$ mean: 0.16, 89% HDI: [-0.06, 0.44]), as attribution rates for outgroup avoidances were very low. These results indicate three group-identity asymmetries in learning parameters. First, priors for zaps were higher for outgroup compared with ingroup target players. Second, learning rates for zaps were higher when observing outgroup bot-players compared with ingroup bot-players. Third, group-level attribution of avoidances was lower when observing outgroup bot-players than ingroup bot-players.

A. Estimated Model Parameters

B. Estimated Learning Pattern

C. Estimated Zap Pattern

Figure 5: Social learning model results We estimated group level parameters of the PLA social learning model. (A) Posterior distribution of the model group-level parameter. 1. Priors include the general bias parameter, and ingroup and outgroup priors. Out-group prior to zap were higher than in-group prior. 2. Learning rates were higher for zaps compared with avoidances and were highest for outgroup zaps. 3. Attribution rates for outgroup avoidance were lower than all other attribution rates. 4. Situational parameters indicate that likelihood to zap increased when stars were far, decreased when the target player was far, and increased when the estimation of the target player’s likelihood to zap was high. (B) Model estimations of the internal learning process about bot-players’ likelihood to zap in the different experimental conditions. Higher learning rates for outgroup zaps, and lower attribution rates for outgroup avoidances contributed to higher estimation of zap probability both for outgroup zappers and avoiders. (C) Model predicted zapping behavior shows a group and behavior effect, like the observed behavioral pattern. Lines in learning curves indicate mean zapping estimation variables, shadows indicate 95% confidence intervals. Boxplots include the median in bold line, interquartile range is represented by the box, minimum and maximum range by the whiskers, and outliers by dots.

4. More analysis on the zap rate measures may be needed. In the section on the model-free analysis (Page 13), the authors examined the aggregated zap rates measures, I think it might be helpful to examine the change of zap rates over time in order to better reflect the “learning mechanisms” (e.g., use the trial index as the predictors, or half-split the trials and compare them against each other), especially when the model-based analysis in the latter section focused on the “learning models”.

We thank the reviewer for this suggestion. We now include a figure and analysis showing the progression of zap probabilities throughout the experiment (Figure 3, see below). As zaps were relatively sparse, we binned the task to four time-bins and examined average zapping frequency in each time-bin. We examined progression of zap frequency in the homogeneous conditions (all-zappers and all-avoiders) and the heterogeneous conditions separately. In both conditions we found the same effects as in the original analysis of zapping behavior in the entire block: a significant group effect (Homogeneous: ($F_{(1, 1798)} = 48.5, p < 0.001, \eta_{\text{partial}}=0.012$), Heterogeneous: ($F_{(1, 1798)} = 29.3, p < 0.001, \eta_{\text{partial}}=0.014$)). In the homogeneous condition we found a significant time effect ($F_{(1, 1798)} = 9.49, p = 0.002, \eta_{\text{partial}}=0.002$), as participants' zapping behavior increased throughout the block, and did not observe a significant behavior effect ($F_{(1, 1798)} = 2.94, p = 0.08, \eta_{\text{partial}}=0.002$), but instead found an interaction between time and behavior ($F_{(2, 1798)} = 5.27, p = 0.021, \eta_{\text{partial}} = 0.001$) as likelihood to zap avoiders did not increase over time. This demonstrates a learning effect on zapping, and this effect is dependent on the players' behavior. In the heterogeneous condition we found a time effect ($F_{(1, 1798)} = 6.23, p = 0.012, \eta_{\text{partial}}=0.003$), and a significant behavior effect ($F_{(2, 1798)} = 4.6, p = 0.03, \eta_{\text{partial}}=0.002$), but did not observe a significant interaction effect ($F_{(2, 1798)} = 0.06, p = 0.79, \eta_{\text{partial}} < 0.0001$), indicating that participants did not differentiate that much between zappers and avoiders in their learning patterns, in line with a group-level attribution effect.

Changes to text

We updated figure 3 to include these learning patterns, and the text in pages 14-15:

We followed this analysis with an exploratory time-bin analysis, calculating zapping rates in four consecutive 25-trial long time bins, to examine how observation of bot-players' behavior throughout the experiment shaped participants' zapping decisions (Figure 3 and tables ST3 ST4 in the supplementary materials). We used mixed-effects linear regression, which included group identity (ingroup/outgroup), bot-player's behavior (zapper/avoider), and a continuous time-bin factor (1-4), and the interactions between these factors as dependent variables, and zapping frequency as dependent variable. We analyzed the homogenous and heterogeneous conditions independently. In both conditions we found a significant group effect (Homogeneous: ($F_{(1, 1798)} = 48.5, p < 0.001, \eta_{\text{partial}}=0.012$), Heterogeneous: ($F_{(1, 1798)} = 29.3, p < 0.001, \eta_{\text{partial}}=0.014$)), in line with the full-block analysis. In the homogeneous condition we found a significant time-bin effect ($F_{(1, 1798)} = 9.49, p = 0.002, \eta_{\text{partial}}=0.002$), as participants' zapping behavior increased throughout the block, and did not observe a significant behavior effect ($F_{(1, 1798)} = 2.94, p = 0.08, \eta_{\text{partial}}=0.002$), but found an interaction between time-bin and behavior ($F_{(2, 1798)} = 5.27, p = 0.021, \eta_{\text{partial}}= 0.001$), as likelihood to zap avoiders did not increase over time whereas likelihood to zap zappers

increased over time. This demonstrates a learning effect on zapping decisions, and this effect is dependent on the players' behavior. In the heterogeneous condition we found a significant time-bin effect ($F(1, 1798) = 6.23, p = 0.012, \eta_{\text{partial}}=0.003$), and a significant behavior effect ($F(2, 1798) = 4.6, p = 0.03, \eta_{\text{partial}}=0.002$), but did not observe a significant interaction effect ($F(2, 1798) = 0.06, p = 0.79, \eta_{\text{partial}} < 0.0001$), indicating that participants did not differentiate that much between zappers and avoiders in their learning trajectories, in line with a group-level attribution effect.

Figure 3: Group-identity and bot-players' behavior effects on participants' zapping behaviour experimental outcomes. Zapping rates throughout the experimental block (left panels) and progression of zapping rates throughout the blocks (right panels) in the homogeneous (A) and the heterogeneous (B) conditions. Participants zapped outgroup bot-players more than neutral and ingroup bot-players, indicating a consistent intergroup bias. Participants were also more likely to zap bot-players that displayed zapping behavior than zap-avoiders and increased their likelihood to zap zappers over time, indicating learning effect on behavior. In the heterogeneous condition participants showed reduced differentiation between avoiders and zappers in the progression of zapping rates, indicating a group-level attribution effect. Lines in learning curves indicate mean zapping frequencies, shadows indicate 95% confidence intervals of the mean. Boxplots include the mean by light circles, median in bold line, interquartile range is represented by the box, minimum and maximum range by the whiskers, and outliers by black dots.

5. Detailed information on the star-harvest game should be provided in the methods section. For example, how many trials were in the experiment? What is the proportion of the zappers choosing to zap the players? These were important information that need to be mentioned in the methods section.

We revised the description of the task, bot in the methods and in the introduction and results sections .

For example, we now explain the way the algorithms governing the behavior of both zappers and zap-avoiders bot-players. The design of the algorithms governing the bot players was inspired by the biases observed in participants when designing the bot-players' algorithms (Hertz 2021)(See figure below, now in the supplementary materials), and was carried to capture levels of players' competitiveness. All bots were programmed to first check on every turn whether they are the closest player to a star, and if so to move towards the star, thus concluding their turn. Otherwise, zap-avoiders are programmed to move away from other players. Zappers, however, check whether another player is on their path to the star closest to them, and if so, zap this player. Otherwise, zappers were also programmed to move away from other players. Zapping is therefore done in context of competition over a star, and not arbitrarily, and is not very frequent, i.e., does not occur every time the bot has the opportunity to zap. The two algorithms therefore resemble the behavior of star-seeking players with different levels of competitiveness, and not random zappers or random avoiders. While this may not completely overcome that problem of participants questioning the identity of other players, we believe it mitigates this problem to some extent, and allows participants to maintain some level of belief that other players may be human as well.

Changes to text:

Page 6 (Introduction):

To test these hypotheses, we adapted a sequential social dilemma paradigm called the star-harvest game^{42,53}, in which five players collect stars and are allowed to sacrifice a move to zap other players and send them to a time-out zone for three turns (Figure 1A). We employed a computational social learning model to this game to study how trial-by-trial experiences and asymmetric learning processes shape intergroup bias. Participants played the star-harvest game with four other players. Participants were not informed about the identity of the other players. Those players were in fact bot-players and were programmed to manifest one of two behavioral patterns. All bot-players were programmed to display star-seeking behavior, and differed in the way they react when another player was standing between them and a star. One type of bot-player zapped this player, clearing the path for them to collect the star, a type called 'zappers'. The other type of bot-player did not zap, and instead moved away towards other stars, a type called 'avoiders' (see methods for full details).

Page 27 (Methods):

The behavior of the bot-players was governed by algorithms implementing zapper and avoider behavior, as used in a previous study⁴² (Supplementary Figure S1). Both bot-player types were programmed to first check whether they are the closest player to a star, and if so to move towards the star, thus concluding their turn. Otherwise, bot-players were programmed to check whether they are in direct competition with another player for a star. This happens when another player is on their way to a star, i.e., closer to the star closest to them. Avoiders were programmed to move away and seek other stars. Zappers were programmed to zap the competing player if they share the same row or column. Zapping is therefore done in the context of competition over a star, and not arbitrarily, i.e., does not occur every time the bot-player *can* zap another player. The two algorithms capture the behavior of star-seeking players with different levels of competitiveness.

Procedure

Participants were directed to the experimental task website and received instructions about the game play. They were then told that they would play the game with four other players, but were not explicitly informed about the players' identity, and whether they were humans or bots. They then continued to choose their avatar color, and to the task. The task lasted 100 turns; each turn included actions from all five players (unless they were in the time-out zone). The order of players' moves on each turn was kept constant. We collected the location of players and stars on each turn and move, and the actions carried by the players. The game lasted 12 minutes on average.

At the end of the task, we included three items to examine the effects of the group manipulation and the participants' experience in the task. First, we asked them to distribute ten extra stars between the four players with whom they played. All ten stars had to be allocated. Second, we asked participants to rate the intentions behind the behavior of the other players. We asked them to rate harmful intent ('prevent others from gaining stars') and selfish intent ('gain more stars'), as previous research found intergroup effects on such judgments⁷⁴. Analysis and results of the intention items are included in the supplementary materials.

Supplementary materials:

Avoider Bot-Player

Zapper Bot-Player

Figure S1 – Algorithms governing the zapping behavior of the zap-avoiders (top) and zappers (bottom) bot-players.

6. If I understand correctly, the authors used the same learning rate to capture the Group-Attribution effect as well as the learning effect (i.e., Page 9, equations 2 and 3 share the same learning rate [LRIn/Out]). Did the author consider using different learning rates here to distinguish the learning effect and group-attribution effect?

We thank the reviewer for this comment. We implemented it as part of the revised formulation of our model, and found that it gave a more stable fitting and therefore we replaced our original formulation with this one. We also agree that this formulation is more straightforward and easier to understand. See our detailed changes to text in previous comments.

Minor:

1. The supplementary information was not well organized. The format of the tables was not matched.

We revised the supplementary information and organized the tables.

2. In Figure 2, the meaning of the shaded area was not define.

We revised all figures with explanations of the different graph elements.

References:

- Aguinis, Herman, Isabel Villamor, and Ravi S. Ramani. 2021. "MTurk Research: Review and Recommendations." *Journal of Management* 47 (4): 823–37.
- Allidina, Suraiya, and William A. Cunningham. 2021. "Avoidance Begets Avoidance: A Computational Account of Negative Stereotype Persistence." *Journal of Experimental Psychology. General* 150 (10): 2078–99.
- Boyer, Pascal, Rengin Firat, and Florian van Leeuwen. 2015. "Safety, Threat, and Stress in Intergroup Relations: A Coalitional Index Model." *Perspectives on Psychological Science: A Journal of the Association for Psychological Science* 10 (4): 434–50.
- De Dreu, Carsten K. W., Jörg Gross, and Angelo Romano. 2023. "Group Formation and the Evolution of Human Social Organization." *Perspectives on Psychological Science: A Journal of the Association for Psychological Science*, July, 17456916231179156.
- Hertz, Uri. 2021. "Learning How to Behave: Cognitive Learning Processes Account for Asymmetries in Adaptation to Social Norms." *Proceedings. Biological Sciences / The Royal Society* 288 (1952): 20210293.
- Hertz, Uri, Stefano Palminteri, Silvia Brunetti, Cecilie Olesen, Chris D. Frith, and Bahador Bahrami. 2017. "Neural Computations Underpinning the Strategic Management of Influence in Advice Giving." *Nature Communications* 8 (1): 2191.
- Hertz, Uri, Evangelia Tyropoulou, Cecilie Traberg, and Bahador Bahrami. 2020. "Self-Competence Increases the Willingness to Pay for Social Influence." *Scientific Reports* 10 (1): 17813.
- Kardosh, Rasha, Asael Y. Sklar, Alon Goldstein, Yoni Pertzov, and Ran R. Hassin. 2022. "Minority Saliency and the Overestimation of Individuals from Minority Groups in Perception and Memory." *Proceedings of the National Academy of Sciences of the United States of America* 119 (12): e2116884119.
- Levy, Jonathan, Moran Inlus, Shafiq Masalha, Abraham Goldstein, and Ruth Feldman. 2022. "Dialogue Intervention for Youth amidst Intractable Conflict Attenuates Neural Prejudice Response and Promotes Adults' Peacemaking." *PNAS Nexus* 1 (5): gac236.
- Shin, Yeon Soon, and Yael Niv. 2021. "Biased Evaluations Emerge from Inferring Hidden Causes." *Nature Human Behaviour* 5 (9): 1180–89.
- Summerville, Amy, and Christopher R. Chartier. 2013. "Pseudo-Dyadic 'Interaction' on Amazon's Mechanical Turk." *Behavior Research Methods* 45 (1): 116–24.
- Tajfel, Henri, M. G. Billig, R. P. Bundy, and Claude Flament. 1971. "Social Categorization and Intergroup Behaviour." *European Journal of Social Psychology* 1 (2): 149–78.
- Thomas, Kyle A., and Scott Clifford. 2017. "Validity and Mechanical Turk: An Assessment of Exclusion Methods and Interactive Experiments." *Computers in Human Behavior* 77: 184–97.
- Xu, X., X. Zuo, X. Wang, and S. Han. 2009. "Do You Feel My Pain? Racial Group Membership Modulates Empathic Neural Responses." *Journal of Neuroscience* 29 (26): 8525–29.

Zaatri, Silina, Idan M. Aderka, and Uri Hertz. 2022. "Blend in or Stand out: Social Anxiety Levels Shape Information-Sharing Strategies." *Proceedings. Biological Sciences / The Royal Society* 289 (1975): 20220476.

29th Nov 23

Dear Orit,

Your manuscript titled "Asymmetric cognitive learning mechanisms underlying the persistence of intergroup bias" has now been seen by our reviewers, whose comments appear below. In light of their advice I am delighted to say that we are happy, in principle, to publish a suitably revised version in Communications Psychology under the open access CC BY license (Creative Commons Attribution v4.0 International License).

We therefore invite you to revise your paper one last time to address the remaining concerns of our reviewers and a list of editorial requests. You will see that the concerns voiced by Reviewer #2 call for greater nuance in how you interpret participant behaviour. At the same time we ask that you edit your manuscript to comply with our format requirements and to maximise the accessibility and therefore the impact of your work.

EDITORIAL REQUESTS:

SUBMISSION INFORMATION:

OPEN ACCESS:

Communications Psychology is a fully open access journal. Articles are made freely accessible on publication under a [CC BY license](http://creativecommons.org/licenses/by/4.0) (Creative Commons Attribution 4.0 International License). This license allows maximum dissemination and re-use of open access materials and is preferred by many research funding bodies.

For further information about article processing charges, open access funding, and advice and support from Nature Research, please visit <https://www.nature.com/commspsychol/article-processing-charges>

At acceptance, you will be provided with instructions for completing this CC BY license on behalf of all authors. This grants us the necessary permissions to publish your paper. Additionally, you will be asked to declare that all required third party permissions have been obtained, and to provide billing

information in order to pay the article-processing charge (APC).

* **DATA AVAILABILITY:**

[link redacted]

Best regards,

Marike

Marike Schiffer, PhD
Chief Editor
Communications Psychology

REVIEWERS' COMMENTS:

Reviewer #1 (Remarks to the Author):

The authors have done an excellent job with their revision of this manuscript. Their response letter and changes to the manuscript have fully addressed the issues I raised in my initial review. I thank the authors for the thoroughness and thoughtfulness of their responses. I think that the paper has been substantially improved by the changes the authors made in response to my and the other reviewer's comments, and is now suitable for publication.

Best,
Mark Thornton

Reviewer #2 (Remarks to the Author):

I want to thank the authors for all their time and efforts in addressing my concerns, and I am also happy that the new model reached a better fit!

I only have a few remarks, mostly revolving around the explanation of the parameters. If I understand correctly, using the new formula they found the attribution parameter for the avoidance behavior is smaller for the outgroup compared to the ingroup, suggesting people were less likely to attribute the positive behavior of one out-group player to other out-group players. I think this result may go against the "outgroup homogeneity" hypothesis, as following this hypothesis, we would expect the attribution parameter to be larger for the outgroup compared to the ingroup, regardless of the valence? Therefore, the authors may want to revise this part accordingly. Similarly, in Figure 5B, if we compare the Het_In, Het_Out subplots, it seems that the Het_In showed a larger attribution effect compared to Het_Out, as the zapping behavior towards the zappers and avoiders seems to be more similar in the ingroup condition compared to the outgroup condition?

Minor:

Page 20, "Learning rates were overall higher for zaps than for avoidances within each group, in line with previous works" Lack of reference here.

We thank the reviewers for their appreciation of our efforts. We believe that our work improved and benefitted from their thoughtful comments and suggestions.

REVIEWERS' COMMENTS:

Reviewer #1 (Remarks to the Author):

The authors have done an excellent job with their revision of this manuscript. Their response letter and changes to the manuscript have fully addressed the issues I raised in my initial review. I thank the authors for the thoroughness and thoughtfulness of their responses. I think that the paper has been substantially improved by the changes the authors made in response to my and the other reviewer's comments, and is now suitable for publication.

Best,
Mark Thornton

Answer:

Thank you very much.

Reviewer #2 (Remarks to the Author):

I want to thank the authors for all their time and efforts in addressing my concerns, and I am also happy that the new model reached a better fit!

I only have a few remarks, mostly revolving around the explanation of the parameters. If I understand correctly, using the new formula they found the attribution parameter for the avoidance behavior is smaller for the outgroup compared to the ingroup, suggesting people were less likely to attribute the positive behavior of one out-group player to other out-group players. I think this result may go against the "outgroup homogeneity" hypothesis, as following this hypothesis, we would expect the attribution parameter to be larger for the outgroup compared to the ingroup, regardless of the valance? Therefore, the authors may want to revise this part accordingly. Similarly, in Figure 5B, if we compare the Het_In, Het_Out subplots, it seems that the Het_In showed a larger attribution effect compared to Het_Out, as the zapping behavior towards the zappers and avoiders seems to be more similar in the ingroup condition compared to the outgroup condition?

Answer:

We agree that our results do not show complete outgroup homogeneity, but rather an ingroup homogeneity and negatively biased outgroup attribution. We rephrased our interpretation and discussion of these findings.

See highlighted changes in text on pages 6 (Introduction) and 29 (Discussion):

...

Finally, group-level attribution may affect learning about ingroup and outgroup members differently and may operate differently on competitive and cooperative actions. For example, it may be manifested in a higher degree of group-level attribution of competitive actions from one outgroup member to another, making all outgroup members seem more likely to perform competitive acts.

Finally, we found that participants tended to attribute negative behaviors, but not positive behaviors, of one outgroup member to all other outgroup members. For in-group, however, attribution was symmetrical. This was observed both in the model parameters and in the time-bin analysis and post-task star allocation pattern. The symmetrical attribution for in-groups is in line with recent works showing greater learning and more flexible impressions formation about ingroup members ⁶². However, our findings suggest a more complex effect of outgroup homogeneity on intergroup bias than initially predicted, demonstrated by our model's group-level attribution rates. Outgroup homogeneity builds on perceptual bias, where outgroups are less distinguishable from one another. This may be related to differential neural processing of same-race and other-race faces ⁶⁴, which can later lead to differences in social categorization. In another study, racial minorities were perceived as more salient than majorities, indicating a lower sense of variability among other-race faces ⁶⁵. In the current study, participants tended to attribute negative behavior from one outgroup member to another, but not positive behavior. This indicates that in the context of social learning, group-level attribution is not symmetrical but is affected by stereotypes and the content of the actions. People are more ready to view all outgroup members as similarly *competitive*, but not similarly *cooperative*. This effect may attenuate the effect of experience on stereotype change by hampering learners' ability to properly track individual behavior, as it is concurrently confounded by evidence from other individuals. Interestingly, we found symmetrical group-level attribution when learning about ingroup members' behavior, which is more in line with the homogeneity assumption. This symmetrical attribution led to reduced belief about the zapping behavior of ingroup zappers, as the avoidance of the other group members was also attributed to them. Homogeneity assumptions may therefore serve different outcomes in intergroup learning processes. Together with the prior effect and learning rates, group-level attribution contributes to the persistence of intergroup bias.

Minor:

Page 20, "Learning rates were overall higher for zaps than for avoidances within each group, in line with previous works" Lack of reference here.

Answer:

We added the missing reference, thanks for pointing this out.